# How does 4DVar data assimilation affect the vertical representation of mesoscale eddies? A case study with OSSEs using ROMS v3.9

David E. Gwyther[1], Shane R. Keating[2], Colette Kerry[1], and Moninya Roughan[1]

[1]Coastal and Regional Oceanography Lab, School of Biological, Earth and Environmental Sciences, UNSW Sydney, Sydney, NSW, Australia
[2]School of Mathematics and Statistics, UNSW Sydney, Sydney, NSW, Australia

**Correspondence:** David E. Gwyther (david.gwyther@gmail.com)

**Abstract.** Accurate estimates and forecasts of ocean eddies in key regions such as Western Boundary Currents are important for weather and climate, biology, navigation and search and rescue. The dynamic nature of mesoscale eddies requires data assimilation to produce accurate eddy timings and locations in ocean model simulations. However, data assimilating models are rarely assessed below the surface due to a paucity of observations, hence it is not clear how data assimilation impacts the subsurface eddy structure. Here, we use a suite of Observing System Simulation Experiments to show how the subsurface representation of eddies is changed within data assimilating simulations even when assimilating nearby observations. We examine in detail two possible manifestations of how the data assimilation process impacts 3-dimensional eddy structure, namely, by producing overly active baroclinic instability and through inaccurate vertical mode structure. Therefore in DA simulations, subsurface temperature structures can be too deep and too warm, particularly in dynamic eddy features. Our analyses demonstrate the need for further basic research in ocean data assimilation methodologies to improve representation of subsurface ocean structure.

## 1 Introduction

Mesoscale ocean eddies are energetic, $\mathcal{O}$(10-100) km wide, rotating circulations with a typical lifespan greater than a month (Gill et al., 1974). Eddies are found ubiquitously throughout the ocean (Chelton et al., 2011), particularly in dynamic current regimes such as where Western Boundary Currents (WBCs) meander and lose coherency (e.g. Mata et al., 2006). Due to their size and lifespan, mesoscale eddies and their peripheral ring of fluid (see Wang et al., 2016; Abernathey and Haller, 2018; Denes et al., 2022) can potentially transport significant quantities of heat and salt (Dong et al., 2014) and therefore water-masses (Zhang et al., 2014) across different regions; provide mixing (e.g. Klocker and Abernathey, 2014); and, they deliver nutrients for biological processes (e.g. McGillicuddy et al., 1998). They can also heavily impact cross shelf exchange with coastal seas (Brink, 2016; Malan et al., 2020), the poleward transport of ocean warming-driving heat (Li et al., 2022a) and marine heatwaves (Elzahaby et al., 2021), which thus influence local Blue Economies (e.g. Li et al., 2017). Data assimilation (DA) simulations, which use observations to produce an optimised estimate of the ocean state, are the obvious choice for producing an accurate representation of eddy location and timing, and thus, predictability — all of which are important due to the myriad impacts of mesoscale eddies.

While DA simulations can place eddies at the correct location and time, they have been shown to be hampered in their subsurface representation. For example, Pilo et al. (2018) considered the impact of DA (specifically using an Ensemble Optimal Interpolation method) on eddy representation and found that model adjustments were forcing nonphysical vertical velocities, temperature and salinity. While this particular DA artefact may not impact all DA systems and methods, other studies have shown that the mean subsurface state (e.g. temperature and velocities) is poorly estimated, even with assimilation of subsurface observations (e.g. Zavala-Garay et al., 2012; de Paula et al., 2021; Gwyther et al., 2022).

Most studies that assess the performance of the global observing system or operational DA models do so by comparison against surface observations. This is due to the broad and detailed surface datasets obtained in the satellite era and by the relative difficulty in obtaining non-sparse subsurface datasets. The majority of modern subsurface observing systems include Argo floats, supplemented by repeat expendable bathythermograph (XBT) lines and more recently autonomous glider deployments. However, even with the rapid increase in subsurface profiles from Argo deployments (e.g. 2 million temperature and salinity profiles between 1999 and September 2018; Wong et al., 2020), these datasets are sparse in their spatial distribution, deployed irregularly at inconsistent locations, and drift with ocean currents. All of these issues result in a focus on correctly representing surface conditions in models, with assumptions made of geostrophic balance and accurate extrapolation from limited subsurface observations.

A clear limitation in assessing subsurface (and eddy) representation is the lack of (withheld) observations with which the DA simulation can be compared to. A workaround to this problem are Observing System Simulation Experiments (OSSEs), which do not have the requirement of a withheld dataset to compare against. OSSEs are a type of DA experiment in which the observations to be assimilated are extracted from a free-running simulation with the addition of realistic errors (Halliwell et al., 2014). This allows comparison of the OSSE against the free-running reference simulation and a better assessment of the efficacy of the DA system and the observing platform. OSSEs have been used previously for planning and assessment of future observational systems and deployments on near-global (e.g. Schiller et al., 2004; Gasparin et al., 2019; Oke and Schiller, 2007; Ballabrera-Poy et al., 2007; Halliwell et al., 2017) and regional (e.g. Melet et al., 2012) scales, as well as exploring how different observation types improve representation of ocean characteristics in DA systems (e.g. Halliwell et al., 2015; Gwyther et al., 2022). To date, we are unaware of any study that has used OSSEs to investigate the impact of different observation types on vertical subsurface ocean representation in a dynamic eddy field.

We examine the subsurface representation of mesoscale eddies in the East Australian Current (EAC), the WBC of the South Pacific Gyre, in a high-spatial resolution 4DVar DA simulation. The simulation study area (Fig.1a, inset) encompasses the EAC and associated eddy field, consisting of anticyclonic and cyclonic eddies. An example anticyclonic eddy from a free-running simulation (see below) is shown in Fig.1a (boxed region), with the vertical structure of the respective temperature field at $0\,\mathrm{m}$, $250\,\mathrm{m}$, $500\,\mathrm{m}$, $1000\,\mathrm{m}$ and $1500\,\mathrm{m}$ shown in Fig.1b. The vertical representation of eddies in the EAC has seen some limited research suggesting that model representation is poor, specifically in DA simulations: An eddy case study simulated by Oke and Griffin (2011) showed the large eddy exhibited anomalous vertical structure that was too deep with an exaggerated tilt (Roughan et al., 2017) however it is still an open question of why, e.g. if this is an unphysical artefact of the DA process. In a high-resolution model of the EAC system, Kerry and Roughan (2020) showed that the free-running simulation

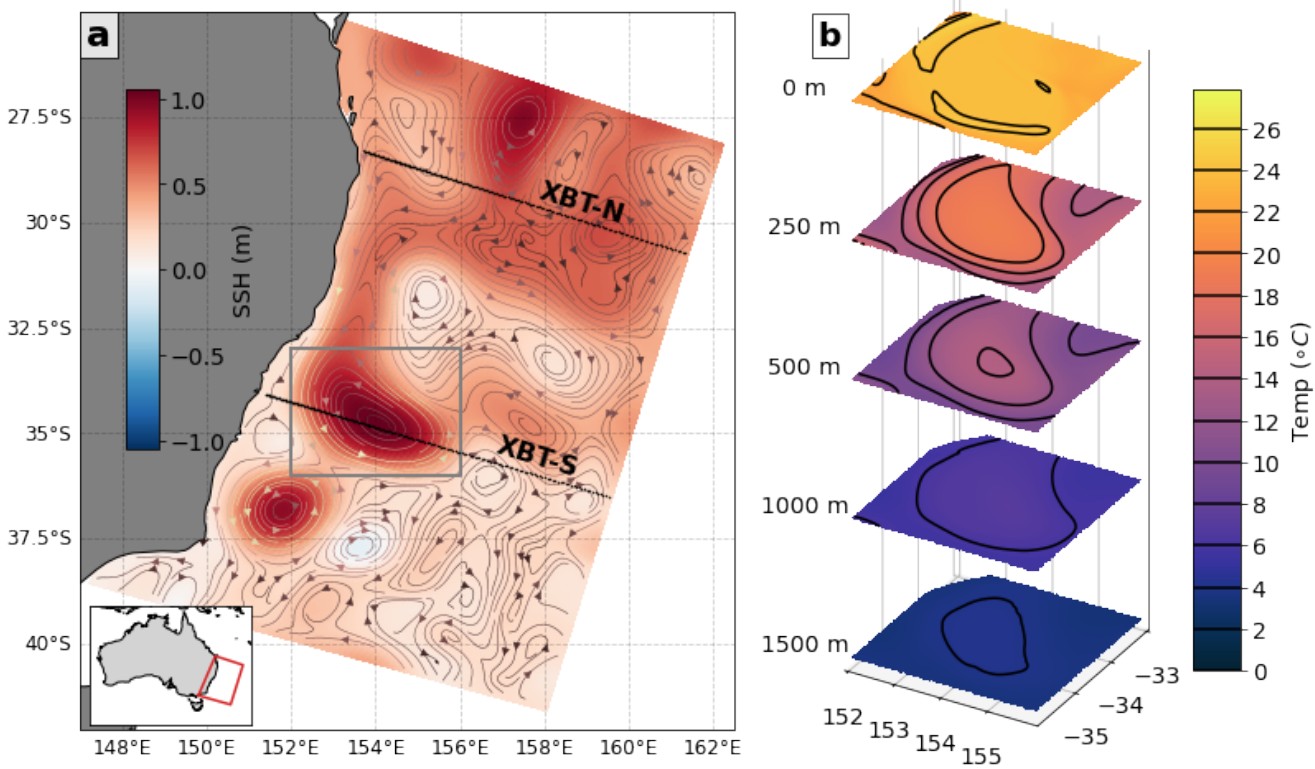

**Figure 1.** (a) The East Australian Current region is shown, with lines of sub-surface observations marked in the north (XBT-N) and south (XBT-S) of the domain (black lines). The colour map shows the mean model sea surface height (SSH) during the 6-12 March 2012, with streamlines calculated from surface velocities marked as vectors. The study region is shown as an inset on the east coast of Australia. The grey box shows the region in focus in panel (b), which shows the vertical structure of temperature at various levels (0 m, 250 m, 500 m, 1000 m, and 1500 m) for a region focussing on an anti-cyclonic (warm core) eddy. Black contours of temperature are shown in (b), with intervals marked in the colour bar.

provided a good representation of 3-dimensional ocean structure. However, when the same model was used in a 4DVar DA configuration, 3-dimensional eddy structure was well represented only in the vicinity of subsurface observations; eddies not located near subsurface observations extended too deep (Siripatana et al., 2020). Gwyther et al. (2022) showed the same for integrated upper ocean heat content, which could be relatively well represented in the vicinity of subsurface observations, but was otherwise poorly represented. Understanding how deficiencies manifest in 3-dimensional eddy representation in DA

systems is required in order to improve predictability in eddy-rich regions such as the East Australian Current.

Using OSSEs, this study explores the subsurface structure of eddies in a series of 4DVar experiments. We diagnose the physical mechanisms by which the vertical representation of eddies is altered as a result in DA simulations. In particular, we focus on two manifestations of this impact: firstly how the model represents the instability that generates eddies, and secondly the cascade of energy through the vertical (baroclinic) modes (e.g. Smith and Vallis, 2001). In Section 2, we introduce the

free running model and DA configuration used in these OSSEs. In Section 3, we present results showing the representation of subsurface conditions and eddy characteristics including case studies of two eddies. In Section 4, we discuss potential mechanisms, including energy conversion and vertical energy distribution, that are hindering more accurate eddy representation and discuss how these limitations are manifest in the vertical ocean dynamics.

## 2 Methods

### 2.1 The Numerical Ocean Model


The Regional Ocean Modeling System (ROMS v3.9 ROMS/TOMS Framework: Mar 3, 2020) is a 3-D finite-difference model solving the primitive equations on a horizontal grid with a terrain following vertical coordinate (Shchepetkin and McWilliams, 2005). The model application used here focusses on the EAC and has been used in several previous studies (e.g. Kerry et al., 2016; Rocha et al., 2019; Siripatana et al., 2020; Li et al., 2021, 2022b; Gwyther et al., 2022). Along the coastline, the model

domain extends from $27°$ S – $38°$ S and over $\sim 700$ km offshore (Fig. 1a). Bathymetry is sourced from the Geoscience Australia 50 m multibeam survey (Whiteway et al., 2009). The grid discretisation has a spatial resolution of 2.5 km to 6 km, linearly increasing in the off-shelf direction, and is rotated $20°$ clockwise from North to approximately align the model grid with the along-continental shelf and off-continental shelf directions. There are 30 vertical $s$-coordinate layers, with the sigmoidal distribution tuned for finer spacing and resolution in the surface layer.

The EAC model application is used with two different configurations, a free-running simulation and the DA configuration. The free-running simulation uses lateral forcing conditions of currents, temperature and salinity from BRAN2020 (Chamberlain et al., 2021, 2020 version of the Bluelink Reanalysis) and daily surface forcing conditions from BARRA-R (Bureau of Meteorology Atmospheric high-resolution Regional Reanalysis for Australia; Su et al., 2019). We refer to this free-running configuration as the 'Ref state'.

The DA configuration used in these OSSEs is an Incremental Strong Constraint 4-Dimensional Variational scheme (IS4D-VAR; e.g. Moore et al., 2011), which has been applied to the EAC region previously (Kerry et al., 2016, 2018). This 4D-Var scheme considers the difference between a free-running forecast and observations (each with associated error fields) over an assimilation window (in our case, 5 days). Adjusted initial and boundary conditions are then generated, such that a new analysis simulation, using these adjusted forcing conditions, has minimised differences (in a least-squares sense) between the analysis

simulation and the observations. The assimilation cycle then increments forward using the previous analysis to initialise a new forecast, and the process repeats.

The DA configuration uses lateral forcing conditions from BRAN2020 and surface forcing conditions from a bulk flux formulation (Fairall et al., 1996) with daily atmospheric conditions from the Australian Bureau of Meteorology's ACCESS reanalysis (Puri et al., 2013). Vertical and horizontal mixing parameters have also been modified between the free-run and

data-assimilating configurations. The different surface forcing conditions and mixing parameters in the assimilating and free-running configurations (see Table C1) are appropriate, as they lead to a source of error that the DA system must reduce, as required in an OSSE. Both the free-running and DA simulations are performed over the period November 2011–January 2013.

The performance and configuration options of the DA system were extensively tested and were shown to produce relatively low error in estimates and forecasts of the EAC (Kerry et al., 2016). The system uses 14 inner loops with one outer loop, set following testing of how many loops were required to achieve acceptable reduction in the cost function (see Gwyther et al., 2022; Kerry et al., 2016). The background error covariances are static and computed by factorisation based on Weaver and Courtier (2001), as described in detail in Kerry et al. (2016).

## 2.2 Observing System Simulation Experiments

OSSEs compare a free-running Reference simulation (the 'Ref state') against data-constrained simulations, where the data to be assimilated is sourced from the Ref state with the addition of errors. The OSSE that is simulating the same period as the Ref state is perturbed to introduce error and initiate divergent evolution through the use of different initial conditions. These initial conditions are similar to those used to initialise the Ref state but are extracted from a point 8 days later (the OSSE begins at 2 December 2011 with conditions from 10 December 2011). This offset is chosen so as to fairly test the DA system Gwyther et al. (see 2022, for further information about this choice of perturbation)

The assimilation of the synthetic observations, which are selected to represent a chosen observation type, location and time, should then converge the resulting analyses towards the Ref state. Comparing the OSSE to the Ref state will show the improvement in the data-constrained reanalysis for the synthetic observation platform tested in each OSSE. For a more detailed description of the procedure, readers are directed to Gwyther et al. (2022) and the schematic outlining the process in their Fig. 2.

The Ref state to which the OSSE is compared should be quasi-realistic of the true ocean and, as a result, the impact of assimilating synthetic observations into the OSSEs should translate to the real ocean. Our Ref state simulation has been rigorously shown to produce an accurate representation of the EAC, including eddy field structure (Kerry et al., 2016; Li et al., 2021), EAC separation latitude (Kerry and Roughan, 2020) as well as long-term conditions (Li et al., 2021). The OSSEs should also be simulated with a sufficient amount of difference to the Ref state such that the ocean state will tend to evolve differently to the Ref state. These differences could arise from different initialisation, model parameterisations, grid resolution and different forcing (Halliwell et al., 2014). We employed a 'fraternal twin' approach for our OSSEs, where we use different initialisation conditions, different model parameterisations (e.g. vertical mixing parameters) and different boundary forcing conditions between the free-running and DA simulations. However, it is also important to ensure that the different configuration of the Ref state and OSSEs (e.g. in this case, surface forcing and some mixing parameters) do not cause such an impact as to introduce a large long-term bias. To assess this, a 'baseline' experiment was conducted using the OSSE configuration, but without assimilating any observations. Comparison of this against the Ref state showed a warm bias in the surface waters, which is likely to be corrected by assimilating SST. More importantly, there is no strong bias in the subsurface ocean, which would otherwise be difficult to correct with assimilation (see Fig. A1).

In this study, we assess the performance of four OSSEs by comparing against the free-running Ref state. These experiments are designed to test the impact of surface-only observations of SSH and SST as measured by satellites (the 'Surf' OSSE), the additional impact of surface observations with XBT-like subsurface temperature measurements (e.g. Scripps PX30 and PX34 XBT lines) in a long transect in the north of the domain (the 'XBT-N' OSSE; see Fig. 1a for transect location), the same

surface observations together with an XBT-like transect of subsurface temperature measurements in the south of the domain (the 'XBT-S' OSSE; see 1a for transect location), and lastly, the surface observations together with the transect of subsurface temperature observations in the north and south of the domain (the 'XBT-N+S' OSSE). Example coverage from SSH, XBT and SST are shown in Fig. B1. Some configuration details and differences between the Ref state and the OSSEs are given in Table C1. Otherwise, we direct readers to Gwyther et al. (2022), where details of the synthetic observations are given, including how their timing and locations are sourced from satellite observations, and the applied observation errors. Background error covariances are set following Kerry et al. (2016), and the reader is directed there for details.

**Table 1.** The experimental configurations of the Ref state and OSSEs are shown, and details of the synthetic observations. Grey fill indicates that the item is not applicable for the Ref state.

| Experiment | Model configuration details | Synthetic observations |
|---|---|---|
| Ref state | Free-running simulation of Nov 2011–Jan 2013. Synthetic observations sourced from this simulation. | None |
| Surf | 4D-Var simulation of Nov 2011–Jan 2013, with 'synthetic observations' of SSH and SST from Ref state | Along-track satellite-observed sea surface height altimetry and sea surface temperature. |
| XBT-N | SSH, SST with XBT observations along a northern transect. | XBT profiles to 900 m starting at $\sim 28^\circ$ S. |
| XBT-S | SSH, SST with XBT observations along the southern transect. | XBT profiles to 900 m starting at $\sim 34^\circ$ S. |
| XBT-N+S | SSH, SST with XBT observations along both transects. | XBT profiles to 900 m starting at $\sim 28^\circ$ S and $\sim 34^\circ$ S. |

## 2.3  Analyses

### 2.3.1  Root-Mean-Square (RMS) Error

The Root-Mean Square (RMS) error is calculated as $RMS = \sqrt{\overline{(\hat{X} - X)^2}}$, where the time mean (shown here as $\overline{\phantom{x}}$) is calculated of the squared difference between the reference field $\hat{X}$ (here, extracted from the Ref state) and the quantity in question $X$ (extracted from the OSSE).

### 2.3.2  Thermocline Depth and Mixed Layer Depth

We use two metrics of upper ocean structure to assess the performance of the 4D-Var simulations for representing this region. The mixed layer depth (MLD) is defined following Fiedler (2010) as the depth at which the temperature is $0.5^\circ$C cooler than the SST at each model grid cell (and time), that is,

$$\text{MLD} = \text{Depth}(T = \text{SST} - 0.5^\circ\text{C}). \tag{1}$$

Here, Depth() denotes the first depth below the surface where the argument in the parentheses is met. The 0.5°C temperature offset is chosen following Fiedler (2010), after inspection of the mean vertical temperature profile, which showed a relatively constant mixed layer could be identified with a smaller offset in temperature. While there are more complex and potentially more dynamically meaningful definitions (such as at sharp changes in temperature and salinity with depth), the above definition is adequate for our purpose: a metric that detects the first, relatively large drop in temperature below the surface which can then be compared between the OSSEs and Ref state.

Below the mixed layer, we identify the thermocline as the transition between warm surface waters and cold, deeper water. We use a similar algorithm to the 'maximum slope by difference' method (Fiedler, 2010); however, we have modified it to suit the mean hydrography present in our model results, which have a thermocline that exhibits a weaker slope in the temperature-depth profile and also extends deeper. Consequently, we capture a representative thermocline depth (TCD) with the criterion

$$\text{TCD} = \text{MLD} - 2\left(\text{MLD} - \text{Depth}\left(\frac{dT}{dz} = \frac{dT}{dz}_{max}\right)\right). \tag{2}$$

That is, the thermocline is at a depth that is twice the distance between the bottom of the mixed layer and the depth of the maximum vertical temperature gradient below the bottom of the mixed layer. Again, a more sophisticated estimate could be used for the thermocline depth, but for our purposes, this metric is sufficient to detect the depth at which surface water transitions to deeper water and is applicable for comparison between our experiments.

### 2.3.3   Eddy Kinetic Energy and Energy Conversions

The eddy kinetic energy (EKE) is defined as the kinetic energy for the perturbations in velocity from the long-term mean, such that $\text{EKE} = \frac{1}{2}\left(u'^2 + v'^2\right)$, where $u'$ and $v'$ are the perturbations in time of the zonal and meridional flow from the long-term average, which is here calculated over the full model integration (November 2011 to January 2013). To gain insight into the energetics in the different experiments, we calculate the conversion rates through barotropic and baroclinic instability. Following Kang and Curchitser (2015), the barotropic conversion rate (KmKe) is through the pathway from mean kinetic energy (MKE) to EKE, calculated as

$$\text{KmKe} = \rho_0\left[\overline{u'u'}\frac{\partial \overline{u}}{\partial x} + \overline{u'v'}\frac{\partial \overline{u}}{\partial y} + \overline{v'u'}\frac{\partial \overline{v}}{\partial x} + \overline{v'v'}\frac{\partial \overline{v}}{\partial y}\right], \tag{3}$$

where $\overline{u}$ and $\overline{v}$ are time-mean zonal and meridional velocities, $u'$ and $v'$ are defined as above and $\rho_0 = 1025 \text{ kg m}^{-3}$. The baroclinic conversion rate (PeKe) is via the pathway from eddy potential energy to EKE, and is calculated as

$$\text{PeKe} = -g\overline{\rho'w'}, \tag{4}$$

where the acceleration due to gravity is $g = 9.81\text{m s}^{-2}$, $\rho'$ and $w'$ are the density perturbation and vertical velocity perturbation from the long-term means calculated at each location, respectively, and the overbar represents a time mean of the enclosed quantity. These quantities have been used effectively in the EAC System and other WBCs to explore energy conversions (e.g. Li et al., 2021, 2022a).

## 2.3.4 Normal mode analysis

We will assess the representation of vertical structure of eddies in each OSSE by analysing the normal modes (i.e. barotropic and baroclinic modes) associated with the density profile at the centre of two case study eddies (see Section .3.3). These modes can then show how kinetic energy is partitioned in the vertical. To begin, the velocity can be decomposed into a sum of orthogonal functions or modes $\phi_n(z)$, with each mode having a time-invariant vertical structure (Gill, 1982), such that

$$u(z) = u_0\phi_0(z) + u_1\phi_1(z) + u_2\phi_2(z) + \cdots = \sum_{n=0}^{\infty} u_n\phi_n(z). \tag{5}$$

Here, $\phi_n(z)$ are the barotropic ($n = 0$) and baroclinic ($n = 1, 2, 3, \cdots$) modes, which are then defined as the solutions to the eigenvalue (Sturm-Liouville) problem (Gill, 1982),

$$\frac{d}{dz}\left(\frac{f^2}{N^2(z)}\frac{d\phi_n(z)}{dz}\right) + k_n^2\phi_n(z) = 0, \tag{6}$$

where $f$ is the Coriolis parameter, $N^2(z)$ is the buoyancy frequency and $k_n$ are the deformation wavenumbers (or equivalently, the inverse deformation length scales). Equation 6 is subject to Neumann boundary conditions on the surface and bottom boundaries, $d\phi_n/dz = 0$ on $z = -H, 0$. The numerical method for solving Equation 6 on a discrete vertical grid is described in Appendix B of (Smith, 2007), and achieved with the code linked below.

The normal modes satisfy the orthogonality condition

$$H^{-1}\int_{-H}^{0}\phi_n(z)\phi_m(z)\,dz = \delta_{n,m} \tag{7}$$

where we have chosen to normalise the modes such that $H^{-1}\int_{-H}^{0}\phi_n^2(z)\,dz = 1$. Making use of Equation 5 and Equation 7, it can be shown that the modal amplitudes are

$$u_n = H^{-1}\int_{-H}^{0} u(z)\phi_n(z)\,dz \approx H^{-1}\sum_{k=0}^{K} u(z_k)\phi_n(z_k)\,\Delta z_k, \tag{8}$$

(with a similar expression for $v_n$) where $k = 0, 1, \cdots, K$ is the layer in a finite vertical layer model, and $z_k$ is the depth of layer $k$.

From this we can also derive the modal decomposition of the depth-integrated kinetic energy (KE), defined as,

$$KE = \frac{1}{2}\int_{-H}^{0}\left(|u(z)|^2 + |v(z)|^2\right)\,dz = \frac{H}{2}\sum_{n=0}^{\infty}\left(u_n^2 + v_n^2\right). \tag{9}$$

Note that the factor of $H$ appears in Equation 9 because $KE$ is the depth-integrated kinetic energy, whereas $u_n$ and $v_n$ are calculated using the depth-averaged orthonormality condition (Equation 7).

## 3 Results

### 3.1 Subsurface conditions across the domain

We first explore the representation of the immediate subsurface, through consideration of the mixed layer depth and thermocline depth. The MLD for all OSSEs (Fig.2b-e) is represented as too shallow when compared to the Ref state (Fig.2a), despite the presence of approximately 6 subsurface observations within the ~40 m thick mixed layer. Note that the mean MLD (shown in each panel in Fig.2) in each OSSE still falls within one standard deviation of the mean MLD of the Ref state. The metric we use here to diagnose a proxy of the MLD indicates that all OSSEs have a near-surface vertical temperature structure that display a more rapid decrease in temperature with depth compared to the Ref state. The shallower MLD of all OSSEs indicates that, in these experiments, there is minimal improvement in near-surface temperature structure offered by the assimilation of observations in this region.

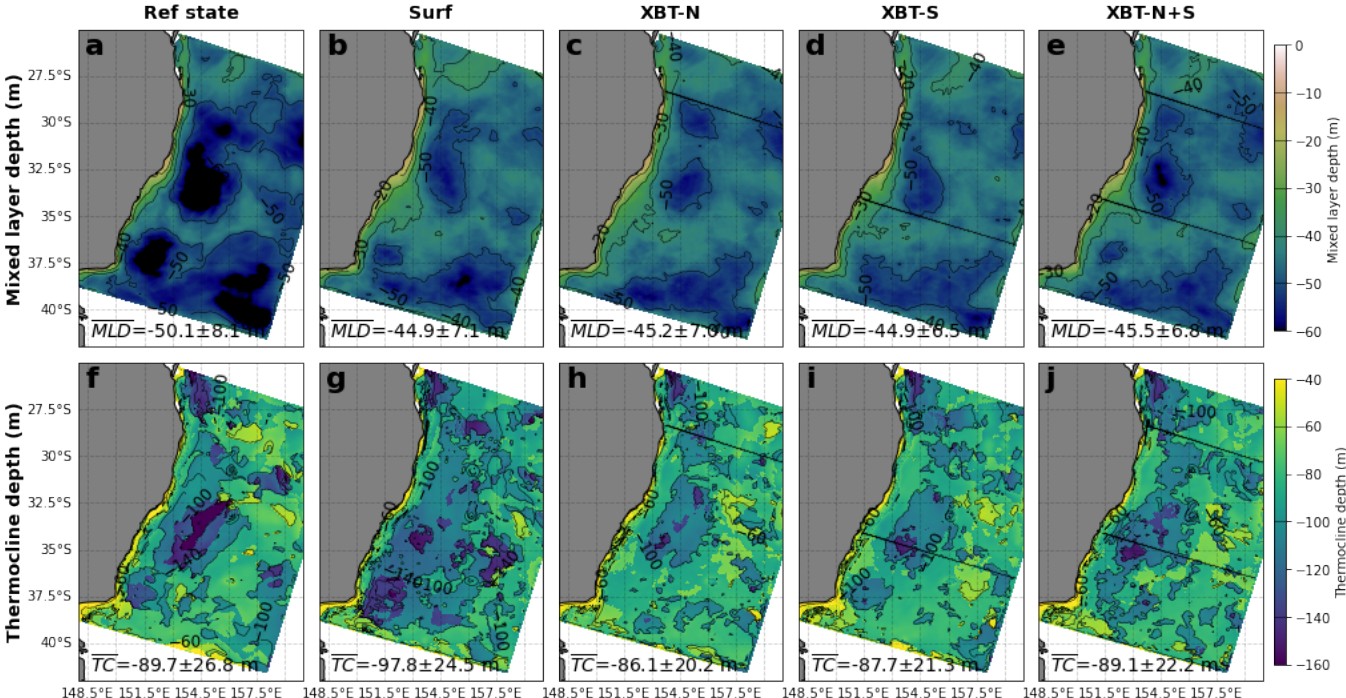

**Figure 2.** The mean mixed layer depth (first row) is shown for the Ref state (a) and each OSSE (b-e). The mixed layer depth is calculated as the depth at which temperature is 0.5°C less than the SST. The thermocline depth (second row) is shown for the Ref state (f) and each OSSE (g-j). The thermocline depth is calculated as the depth equal to twice the distance from the bottom of the mixed layer to the depth of maximum change in temperature with depth. $\overline{MLD}$ and $\overline{TC}$ shows the spatial means and standard deviations of the mixed layer and thermocline depth, respectively. The XBT-N and XBT-S transect locations are marked in respective panels. Axes with latitudes are labelled °S and axes with longitudes are labelled °E.

The thermocline depth, which is deepest in the EAC eddy region (154.5° E, 33–34° S) in the Ref state (Fig.2f), is relatively poorly represented in all OSSEs (Fig.2g-j). The Surf OSSE has a thermocline which is not deep enough in the eddy-rich region (120–140 m deep, compared to over 160 m deep in the Ref state) but too deep for most of the rest of the domain (over 120 m deep, in contrast to the Ref state which outside of the eddy-rich region is 80–100 m deep) as shown in Fig.2g. The presence of subsurface observations improves the spatial pattern of thermocline depth (Fig.2h-j). The thermocline is deepest in the upstream region of the EAC core and in the eddy dominated region between 32.5° S – 36° S. In these deep thermocline regions, all of the XBT OSSEs represent the bottom of the thermocline as too shallow, while the regions of shallower thermocline (outside of the EAC and its eddies) are fairly well represented (and represented considerably better than in the Surf OSSE). This likely points to poor representation of EAC core and eddy vertical structure in all OSSEs.

## 3.2 Eddy Kinetic Energy representation

While surface EKE can be reasonably estimated in the presence of surface (particularly SSH) observations, subsurface EKE, and hence the 3-dimensional structure of eddy variability, has generally poor spatial and temporal representation. We integrate subsurface EKE over two depth ranges, from 0 m to 250 m, and from 250 m to 2000 m, which were chosen to capture the upper (higher energy) and deeper (lower energy) regions of eddy depth structure.

As expected, the mean EKE in the top 250 m is strongest in the EAC eddy region (Fig.3a). The representation of upper ocean EKE in all OSSEs is relatively similar (Fig.3b-e), with the difference from the Ref state being small and having a similar spatial pattern for all OSSEs. At depth, however, the representation of the depth-averaged EKE (Fig.3f) is represented differently in each OSSE. The Surf OSSE overestimates EKE through the highest EKE region (Fig.3g). The OSSEs with a single transect of observations perform better, with lower error in representation of the depth-averaged EKE (Fig.3h-i). The XBT-N+S OSSE has a slightly higher EKE difference than XBT-N or XBT-S but performs better than Surf (cf. Fig.3j and Fig.3g). As discussed in Gwyther et al. (2022), the XBT-N+S OSSE sometimes displays higher error than the single XBT transect OSSEs, which is likely because the DA scheme is forced to minimise errors at both the northern and southern subsurface observation locations. This leads to a degraded fit to either observation transect individually. This has also been demonstrated by others, for example, Siripatana et al. (2020), who found that additional data streams (mooring data and HF radar currents) degraded representation of SSH and SST; and Zhang et al. (2010), who showed that assimilating HF radar currents increased the error in the subsurface temperature forecast.

The performance of each OSSE can be compared in the vertical profiles of EKE averaged over the high EKE variability region (Fig.3k). EKE in the Ref state and all OSSEs is surface-intensified, while EKE in the upper 250 m is relatively well represented in all OSSEs. The EKE in the Ref state continues to decrease with depth, until approximately 1000 m. The XBT-S OSSE matches the Ref state in the monotonic decrease in EKE with depth and provides the best fit, however the decrease with depth is less compared to the Ref state. All other OSSEs display subsurface EKE maxima between ~500–1100 m.

We now consider another measure of the 3-dimensional structure of eddies in these simulations — the along-shelf and across-shelf slope of the temperature fields. The mean temperatures and isotherm slopes at 250 m are shown in Fig.4. This

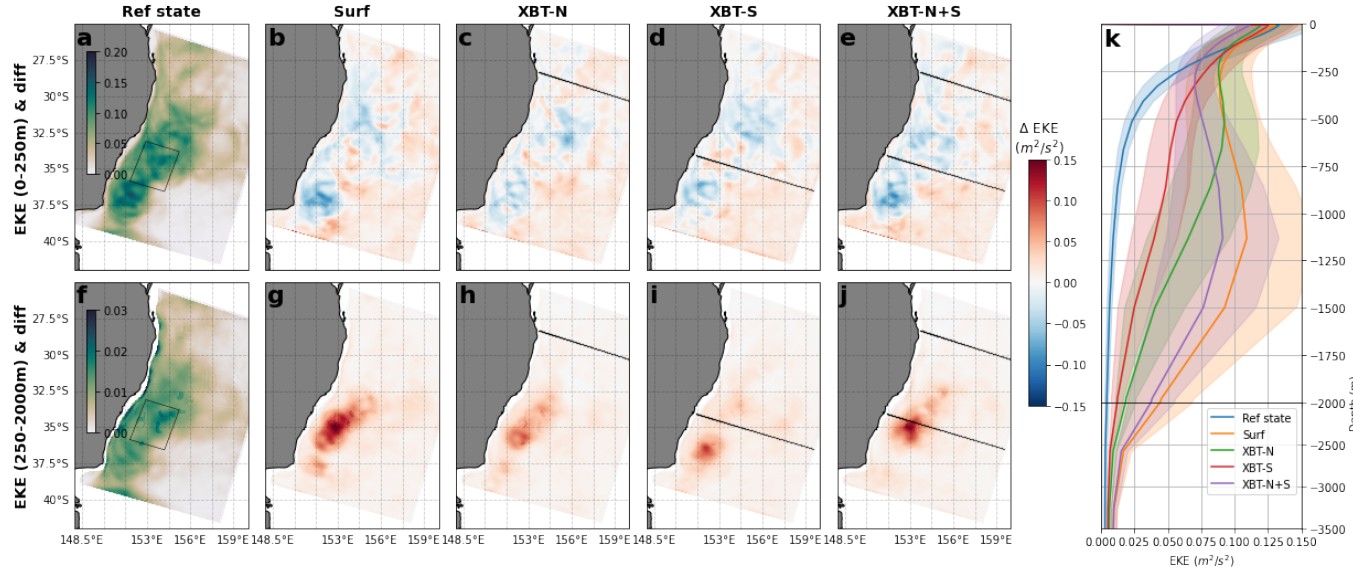

**Figure 3.** The time-mean eddy kinetic energy (EKE) is shown as the average over the top 250 m for (a) the Ref state, and the difference from this for the (b) Surf, (c) XBT-N (d) XBT-S and (e) XBT-N+S OSSEs, and as the average over 250 m –2000 m for (f) the Ref state, and the difference from this for the (g) Surf, (h) XBT-N (i) XBT-S and (j) XBT-N+S OSSEs. In (k), the vertical profile of the EKE, spatially averaged over the high eddy variability region (box shown in panels a and f) is shown for each OSSE. The shaded regions designate a range of plus and minus one standard deviation in spatial mean EKE at that depth, for that OSSE. The XBT-N and XBT-S transect locations are marked in respective panels. Axes with latitudes are labelled °S and axes with longitudes are labelled °E.

depth is chosen as it represents the transition where the OSSEs begin to display enhanced EKE, compared to the Ref state, as shown in Fig.3k.

The presence of sub-surface observations improves the representation of mean temperature (cf. Fig.4b and Fig.4c-e), with
the XBT-S OSSE having the best representation of the higher temperature region at 154.5° E, 33° S. The across-shelf isotherm slope is characterised by a strong, negative slope along the coast, a weaker, broader region of negative slope in the off-shelf region of the western Tasman Sea (e.g. at 151.5° E, 36° S), and, further eastwards (153° E, 36° S), a weak but broad region of positive slope (Fig.4f). These features represent, respectively, the sloping isotherms associated with the southward flowing EAC jet, the western edge of the EAC eddy field, and the northwards return flow. While all OSSEs broadly contain these features
(Fig.4g-j), the representation is most accurate in the presence of subsurface observations in the southern region (XBT-S and XBT-N+S; Fig.4i,j). The relatively high vertical and horizontal spatial resolution of the subsurface observations improves the representation of the isotherm slope near the observation transect, but also the magnitude and distribution of the positive slope in the return flow and the broad, weakly negative slope in the eddy region. However, all OSSEs still represent weaker sloping across-shelf isotherms.

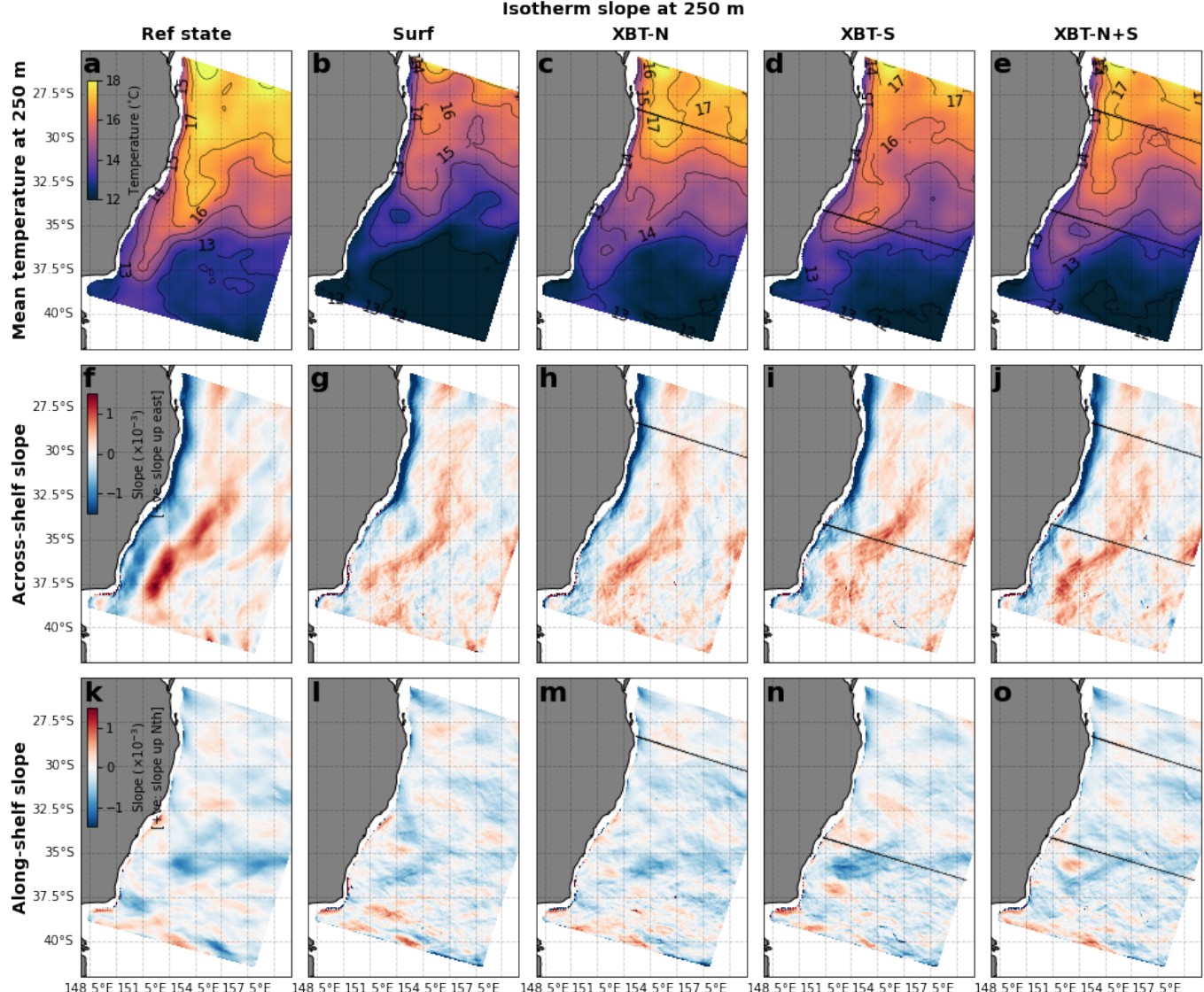

**Figure 4.** The mean temperature at 250 m is shown for (a) the Ref state, and the (b) Surf, (c) XBT-N (d) XBT-S and (e) XBT-N+S OSSEs. The across-shelf isotherm slope at 250 m is shown for the (f) Ref state, and the (g) Surf, (h) XBT-N, (i) XBT-S and (j) XBT-N+S OSSEs. Likewise, the along-shelf isotherm slope at 250 m is shown for the (k) Ref state, and the (l) Surf, (m) XBT-N, (n) XBT-S and (o) XBT-N+S OSSEs. A positive across-shelf slope is upwards sloping towards the east and a positive along-shelf slope is upwards sloping towards the north. The XBT-N and XBT-S transect locations are marked in respective panels. Axes with latitudes are labelled °S and axes with longitudes are labelled °E.

The most notable feature in the Ref state along-shelf slope is the zonal band of negative slope at 35° S, associated with the eastwards extension of the EAC (Fig.4k). This band of negative slope is poorly represented in the Surf OSSE (Fig.4l),

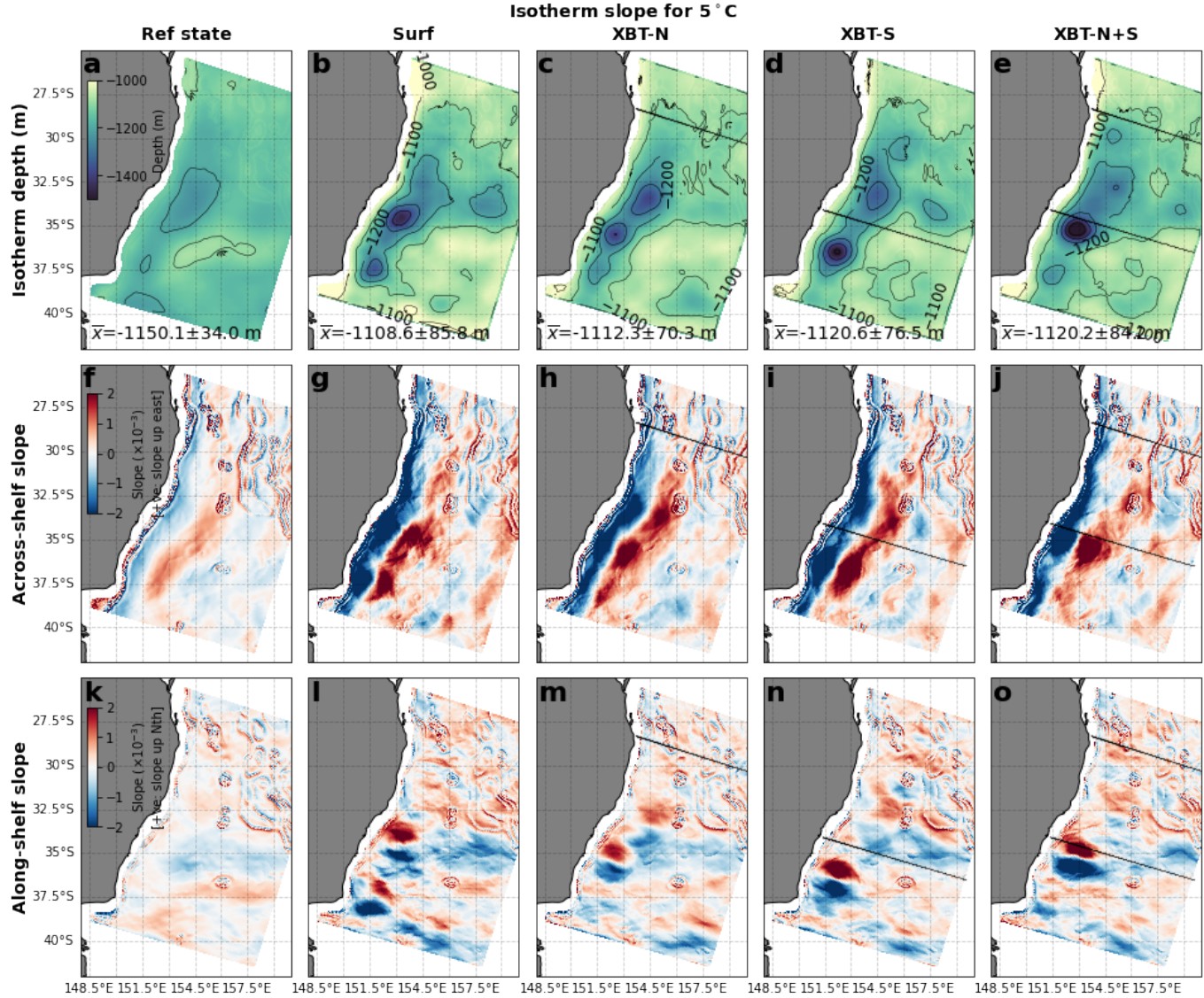

**Figure 5.** The depth of the 5°C isotherm is shown for (a) the Ref state, and the (b) Surf, (c) XBT-N (d) XBT-S and (e) XBT-N+S OSSEs. The across-shelf isotherm slope at the 5°C isotherm is shown for the (f) Ref state, and the (g) Surf, (h) XBT-N, (i) XBT-S and (j) XBT-N+S OSSEs. Likewise, the along-shelf slope of the 5°C isotherm is shown for the (k) Ref state, and the (l) Surf, (m) XBT-N, (n) XBT-S and (o) XBT-N+S OSSEs. A positive across-shelf slope is upwards sloping towards the east and a positive along-shelf slope is upwards sloping towards the north. The XBT-N and XBT-S transect locations are marked in respective panels. Axes with latitudes are labelled °S and axes with longitudes are labelled °E.

improved with subsurface observations (Fig.4m-o) and best represented with the XBT-S observations (Fig.4n). This indicates that subsurface observations through the eddy region are key for improving representation of the EAC eastern extension.

While the 250 m isotherm represented the transition from the upper region of eddies and EKE, the deeper eddy region (which we defined as the region of increased EKE between 250–2000 m) can be captured by the 5°C isotherm. The depth and slope of this isotherm indicates the degree of vertical motion in the eddy field, and the potential to which water in this region could display baroclinicity.

The 5°C isotherm in the Ref state is relatively flat with a mean depth of 1150 m (Fig.5), meaning most of the eddy variability (e.g. upwelling) is above this depth. All OSSEs overestimate the depth of this isotherm in the high EKE region (153° E, 35° S) and underestimate the depth outside of this region (Fig.5b-e). The across-shelf (Fig. 4f) and along-shelf (Fig. 4k) isotherm slopes display features related to the density structure that contributes to driving or maintaining the EAC jet, southward extension, return flow and eastern extension, as outlined above.

Like the isotherm tilting at 250 m (e.g. Fig.4f-j), the slope, both positive and negative, of the 5°C isotherm is over estimated in the across-shelf direction (Fig. 5g-j) and the along-shelf direction (Fig. 5l-o).

## 3.3    Eddy case studies

The above sections have focussed on time-mean metrics, however, a case study analysis is useful for providing insight into the model performance of these dynamic features. We have chosen to focus on two example eddies, where these events represent 'best case' or 'worst case' scenarios for the accurate simulation of subsurface conditions, namely, eddies in the vicinity of or distant from subsurface observations.

### 3.3.1    Case Study A: Eddy on the XBT observations

The first case study considers the vertical structure of an anticyclonic eddy that passed through the XBT-S observation line (centred on the eddy-rich region), averaged over the period 11-March 2012 – 16-March 2012 in the Ref state simulation. This eddy is chosen as one of the two case studies as the co-location with the XBT-S observation line should afford significant improvement in the vertical structure, and hence represent a 'best cast scenario'.

The anticyclonic eddy of case study A is recognisable in the Ref state and OSSEs by the large SSH anomaly centered at approximately 153° E, 35° S (Fig. 6). As each OSSE will have a slightly different simulated eddy, the comparison transect (blue transect lines in maps; Fig. 6a-e) is shifted to pass through the eddy centre and allow a comparison of conditions through the eddy centre.

The Ref state displays deepened isotherms, which are characteristic of anti-cyclonic eddies (Fig. 6f). The representation of vertical temperature in the Surf OSSE is too warm through the eddy core as well as displaying a subsurface lens of doming isotherms (suggesting upwelling) at 250 m depth (Fig. 6g); XBT-N also displays poor representation of temperature, with the upper 500 m being too cold and from 500 m to 1000 m being too warm (Fig. 6h). In contrast, the OSSEs which assimilate the southern XBT observations (XBT-S and XBT-N+S; Fig. 6i-j) display lower error and reasonable vertical temperature representation, though both still suffer from overly warm eddy core water below 1000 m. The upper 500 m in XBT-N+S is also too cold, but not to the same extent as XBT-N (cf. Fig. 6j and h).

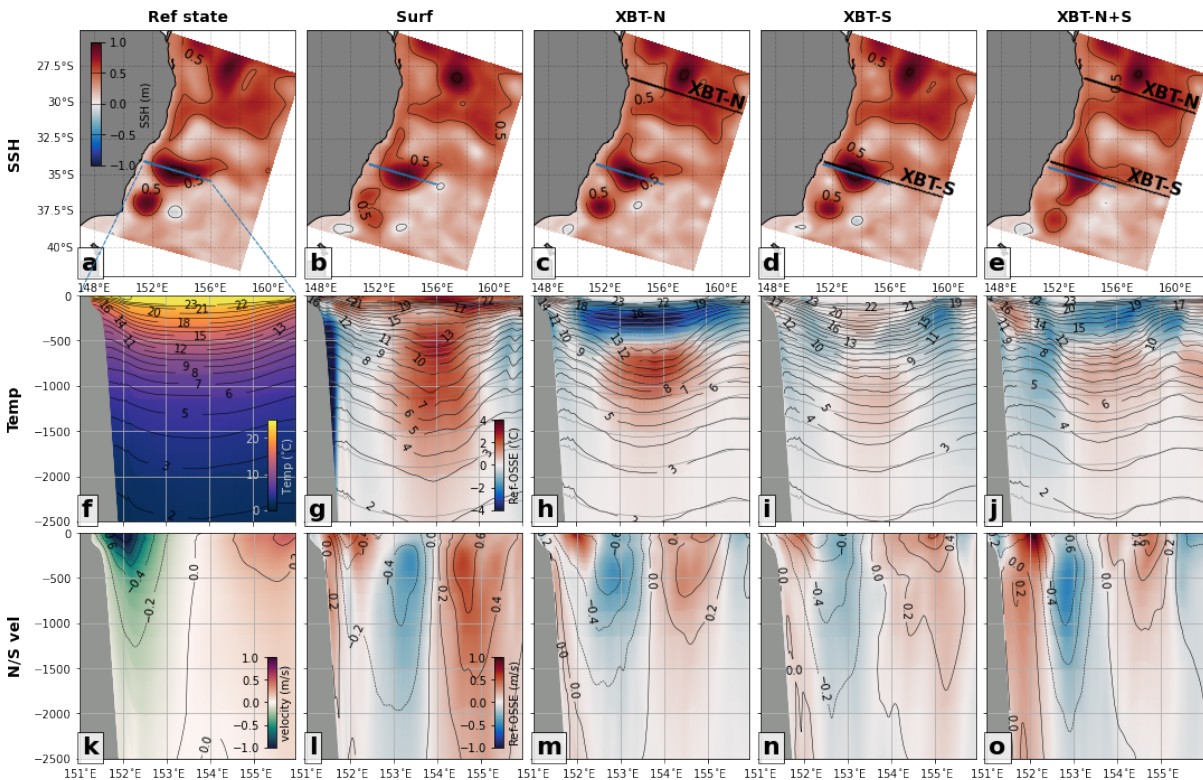

**Figure 6.** The vertical representation of an anticyclonic eddy at ∼153° E, 35° S in the (a) Ref state, is compared to the (b) Surf, (c) XBT-N, (d) XBT-S, and (e) XBT-N+S OSSEs, with the SSH field and vertical profile location marked by a blue line. The vertical temperature profile along the transect is shown for the (f) Ref state, and the (g) Surf, (h) XBT-N, (i) XBT-S and (j) XBT-N+S OSSEs. Likewise, (k-o) Northwards velocity profiles are shown for the same experiments. In the OSSE panels, the difference in temperature and velocity from the Ref state s plotted in colour and contours of the (dark colour) OSSE and (light colour) the respective field in the Ref state are shown for comparison. In panel (a), the blue dashed lines indicate the transect shown in (f). The XBT-N and XBT-S transect locations are marked in respective panels. Axes with latitudes are labelled °S and axes with longitudes are labelled °E.

The North/South velocity is characteristic of a anticyclonic eddy with southward velocity inshore and northward velocity further east (Fig.6k). All OSSEs struggle to represent the velocity field. Velocities in the Surf OSSE are too strong, too deep and horizontally compact (Fig.6l), while the XBT-N and XBT-S OSSEs (Fig.6m-n) have velocity fields closest to the Ref state. Like the Surf OSSE, the XBT-N+S velocity representation is dissimilar to the Ref state, being too narrow and too strong at depth (Fig.6o).

### 3.3.2 Case Study B: Eddy south of the XBT observations

The second case study, case study B, was chosen as it presents a more complex scenario of an anticyclonic and cyclonic eddy pair located at ∼151.5° E – 153.5° E, ∼37.5° S over the same period 11-March 2012 – 16-March 2012. In this case study, no

OSSEs have subsurface observations located closer than ∼300 km — this situation is more akin to a 'worst case scenario' for representing eddy vertical structure in a DA simulation.

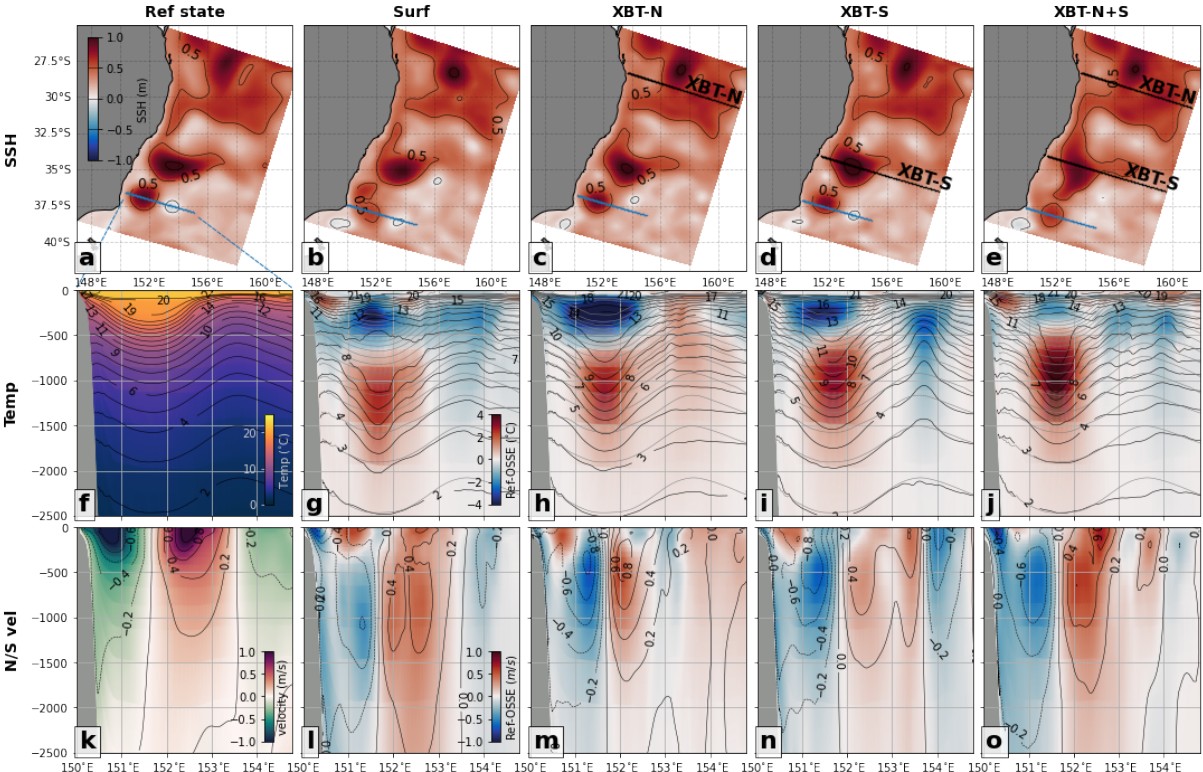

**Figure 7.** The vertical representation of an anticyclonic-cyclonic eddy pair at ∼151.5–153.5° E, 37.5° S in the (a) Ref state, is compared to the (b) Surf, (c) XBT-N, (d) XBT-S, and (e) XBT-N+S OSSEs, with the SSH field and vertical profile location marked by a blue line. The vertical temperature profile along the transect is shown for the (f) Ref state, and the (g) Surf, (h) XBT-N, (i) XBT-S and (j) XBT-N+S OSSEs. Likewise, (k-o) Northwards velocity profiles are shown for the same experiments. In the OSSE panels, the difference in temperature and velocity from the Ref state s plotted in colour and contours of the (dark colour) OSSE and (light colour) the respective field in the Ref state are shown for comparison. In panel (a), the blue dashed lines indicate the transect shown in (f). The XBT-N and XBT-S transect locations are marked in respective panels. Axes with latitudes are labelled °S and axes with longitudes are labelled °E.

The SSH fields in each OSSE differ by varying degrees to the Ref state, with the XBT-N and XBT-N+S simulations containing only the anticyclonic eddy while the Surf and XBT-S OSSEs displaying the signature of both eddies, but with noticeable spatial offsets and SSH magnitudes (Fig. 7 a-e).

The case study B anti-cyclonic eddy has an isothermal core from 100 m to 300 m deep, and a thermal structure typical of an anti-cyclonic eddy with deepened isotherms below that (Fig. 7f). Representation of both of the anticyclonic and cyclonic eddies is poor in all OSSEs. The Surf, XBT-N and XBT-N+S OSSEs display overly deep isotherms (that is, too warm) below 500 m and erroneous uplifting of isotherms, within the anti-cyclonic eddy, between 500 m and 100 m (Fig. 7g,h,j), leading to

a lens-like isothermal layer in the 250–750 m depth range. The XBT-S OSSE is also too cold in the top 500 m and too warm from 500 m to 1500 m (Fig. 7i), but it does not have the uplifted isotherms present (in anti-cyclonic eddy) in the other OSSEs.

The northwards velocity transect through the case study B eddies is a surface-intensified velocity field with relatively symmetric northwards and southwards flow around the western-most eddy core and weaker southward flow on the far side of the more easterly cyclonic eddy; in the vertical, velocity is strongest at the surface and monotonically decreases with depth (Fig. 7k). All OSSEs struggle to represent this, with all representations being subsurface intensified and more spatially complex and asymmetric about the eddy core (Fig. 7l-o) compared to the Ref state. This shows that even with a reasonable surface

impression, subsurface velocity fields of an eddy can be far from representative.

## 3.4  Eddy generation

We have established that the time-mean subsurface conditions are poorly represented especially in the high EKE region, indicating that these DA simulations are struggling to capture the structure of eddies at depth. Indeed, our case studies show that the vertical structure of individual eddies is also poorly represented, whether they are far from or close to observations. We

now consider mechanisms that could be inhibiting a more accurate vertical eddy structure.

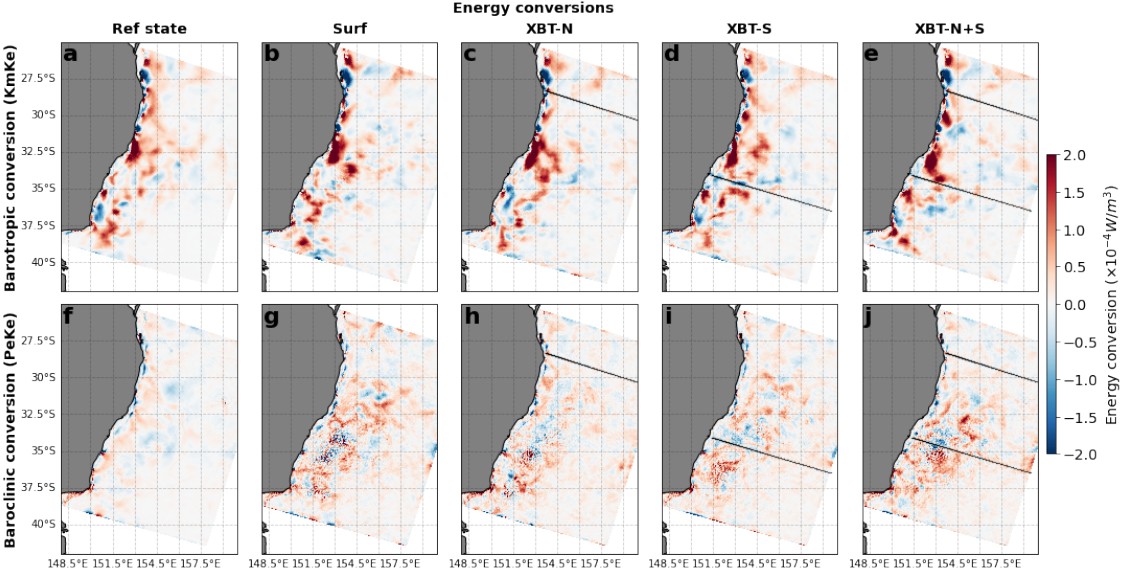

**Figure 8.** The barotropic conversion rate (KmKe) is shown for the (a) Ref state, and (b) Surf, (c) XBT-N, (d) XBT-S, and (e) XBT-N+S OSSEs. The baroclinic conversion rate (PeKe) is shown for the (f) Ref state, and (g) Surf, (h) XBT-N, (i) XBT-S, and (j) XBT-N+S OSSEs. In all panels, a positive value indicates conversion from (for panels a-e) mean kinetic or (for panels f-j) eddy potential energy into eddy kinetic energy. The XBT-N and XBT-S transect locations are marked in respective panels. Axes with latitudes are labelled °S and axes with longitudes are labelled °E.

The barotopic conversion rate (KmKe) captures the energy pathway by which depth-mean horizontal velocity shear instability forms eddies. In contrast, the baroclinic conversion rate (PeKe) captures eddy formation that results from unstable vertical density structure and baroclinic instability. Here we average both quantities over the top 450 m, to capture the region of highest EKE (Fig. 3k).

Comparing KmKe to PeKe we see that barotropic conversion is approximately an order of magnitude larger than baroclinic conversion (cf. Fig. 8a and f), which agrees with results from Li et al. (2021) who suggest KmKe is the dominant mode of eddy production in the EAC. All OSSEs produce a good representation of barotropic conversion rate, with most of the high KmKe hotspots (e.g. 152° E, 32.5° S) being captured (Fig.8b-e). This explains why all OSSEs have a good representation of the time-mean surface EKE field (see Fig. 3b-e).

However, baroclinic production is poorly represented in all OSSEs, with PeKe being too strong and extending too far to the east (Fig. 8g-j). This suggests that the vertical density structure in the eddy field of all OSSEs is such that baroclinic instability is too active and generates too much conversion to eddy kinetic energy.

## 3.5  Normal mode structure

In an effort to explore an alternative manifestation of incorrect eddy representation, we employ a normal mode analysis, whereby the barotropic and baroclinic modes are computed (e.g. Gill, 1982; Wunsch, 1997; Kelly, 2016). This gives insight into how the OSSEs are simulating the vertical partitioning of kinetic energy in each baroclinic mode throughout the water column.

The normal modes are derived from the stratification profile (see Section 2.3.4 and Equation 6) using the numerical implementation described in (Smith, 2007). For case study A (Ref state; Fig.9a), there is a relatively smooth increase in density with depth. In contrast, the Ref state in case study B has a sharper thermocline at $\sim$200 m, weak change in density between $\sim$200 m and 350 m, and below that, a smooth increase in density with depth (Ref state; Fig.10g).

The amplitudes of the first three baroclinic modes, calculated for the centre of the case study A eddy, are shown in Fig. 9b-d. The barotropic mode, $\phi_0$, is normalised to have unity value at all depths, and so we focus attention on the first 3 baroclinic modes $\phi_1$, $\phi_2$ and $\phi_3$. For case study A, XBT-S and XBT-N+S have the most accurate baroclinic mode structure (e.g. RMS values of 0.09–0.2 compared to 0.20–0.46 for Surf and XBT-N; Fig. 9b,c); the other OSSEs have errors in the amplitude with depth of all BC modes (Fig. 9b-d). The improved mode structure in XBT-S and XBT-N+S, particularly of $\phi_2$ (Fig. 9c) likely corresponds to a better representation of the weaker, smoothly sloping thermocline — the other OSSEs display a sharper thermocline at $\sim$200 m.

For case study B, with a more complex density structure (Ref state; Fig. 9e), all OSSEs fail to represent accurate baroclinic mode structure, with higher RMS values. The shape of $\phi_1$ is poorly represented in all OSSEs (RMS errors ranging from 0.3–0.4; Fig. 9f) which likely indicates a failure to capture the increase in density with depth associated with the primary thermocline below $\sim$350 m. The section of near-constant $\phi_1$ amplitude with depth that is present in Surf, XBT-S and XBT-N+S from 250–1250 m likely represents the isothermal lens (Fig. 7g,j). This feature is present in the Ref state but at a much shallower depth (from 100–250 m; Fig. 7f), with the signature showing in $\phi_1$, $\phi_2$ and $\phi_3$ (Fig. 9f-h).

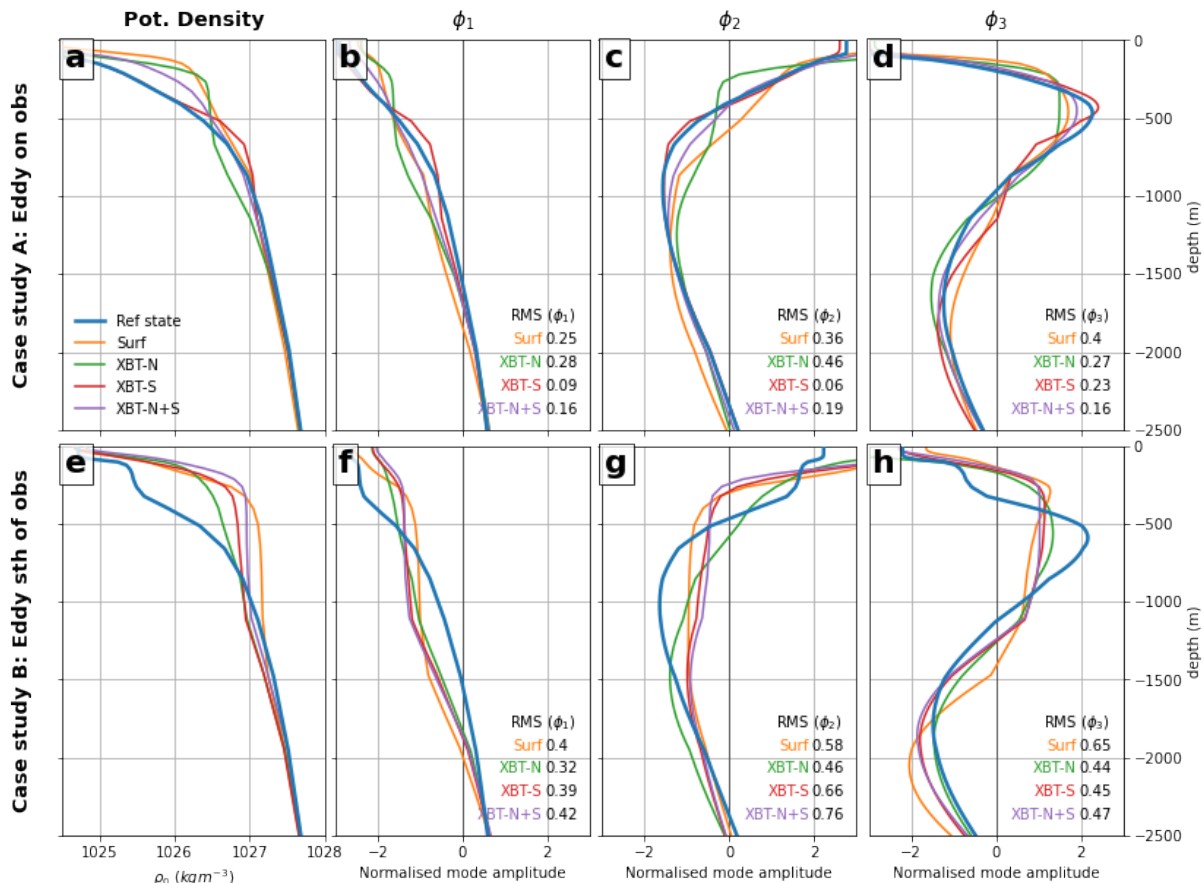

**Figure 9.** For case study A, the (a) potential density profile is shown for each OSSE, and the amplitude of the baroclinic modes (b) $\phi_1$, (c) $\phi_2$ and (d) $\phi_3$ are shown for the Ref state, each OSSE. For case study B, the (e) potential density profile, and baroclinic modes (f) $\phi_1$, (g) $\phi_2$ and (h) $\phi_3$ are shown for each OSSE. The barotropic mode ($\phi_0$) is excluded, as by normalisation it has unity value at all depths. Potential densities are referenced to surface pressure. In panels b-d and f-h, we show the RMS difference of the OSSE mode shape compared to that of the Ref state. Note that the vertical axis has been limited to the top 2500 m .

For case study B, $\phi_2$ is represented differently in all OSSEs compared to the Ref state, having either a portion ($\sim$500–1500 m ) which stays relatively constant in amplitude with depth (Surf, XBT-S and XBT-N+S; Fig. 9g) or by displaying an overly deep maximum (XBT-N; green line in Fig. 9g). The first maximum in $\phi_3$ is too shallow and too weak in all OSSEs (Fig. 9h). The second, deeper maximum in $\phi_3$ is too deep in all OSSEs, but particularly in Surf (Fig. 9h), which shows that all OSSEs represent this eddy as too deep.

Together, these results show that the baroclinic mode structure is poorly represented in all OSSEs, but especially in the absence of nearby observations (case study B). The second baroclinic mode is particularly susceptible to erroneous shape, which corresponds to poor representation of any deviations in the primary thermocline. The third baroclinic mode also has

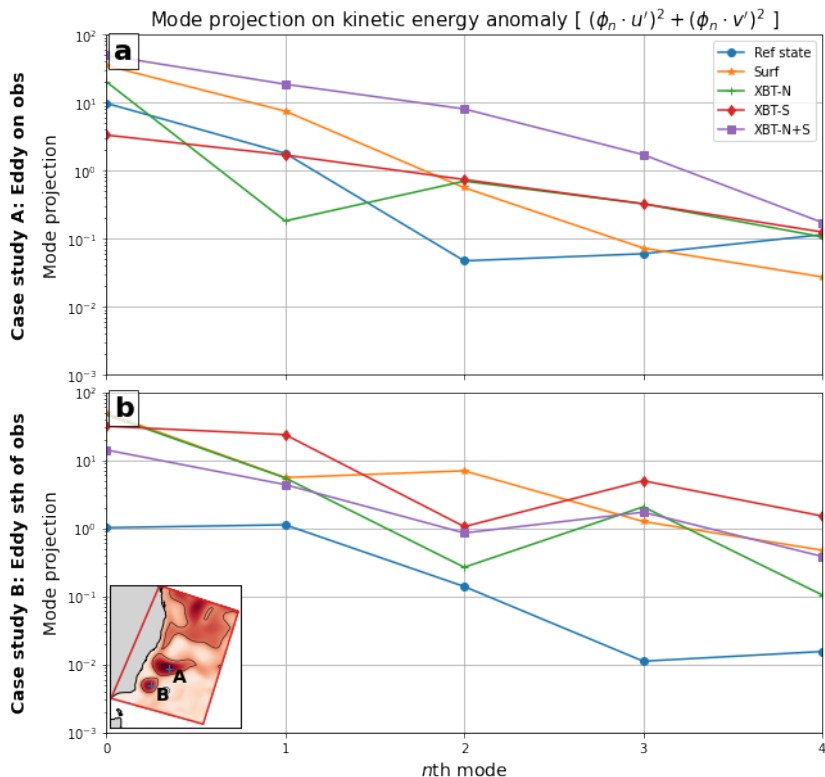

**Figure 10.** The barotropic and baroclinic modes are projected onto the kinetic energy through the squared addition of the mode projections on the zonal and meridional velocity anomalies (see Section 2.3.4). In (a), the value of the $n$th mode projections for case study A are shown, and in (b), the $n$th mode projections for case study B are shown. The locations of case studies A and B are shown in the inset in (b).

poor representation in all the DA experiments, which captures how eddies here are simulated with an overly deep vertical extent.

By projecting these baroclinic modes onto velocity anomalies in the meridional and zonal directions, we can decompose the vertical partitioning of kinetic energy into the components resulting from the different BC modes. This kinetic energy decomposition shows how energy is vertically distributed between the different modes.

In case study A (Fig.10a), the XBT-N and XBT-N+S OSSEs poorly estimate the energy associated with either the barotropic mode and one or more baroclinic modes. The XBT-S OSSE displays a fair estimate of energy in $\phi_1$ and $\phi_4$, but underestimates

$\phi_0$ and overestimates $\phi_2-\phi_3$. The Surf OSSE represents $\phi_3$ well, but overestimates $\phi_0$, $\phi_1$ and $\phi_2$.

At case study B (Fig. 10b) all OSSEs overestimate the energy distribution in the barotropic and baroclinic modes. In particular, $\phi_2$ is best represented by XBT-N, XBT-S and XBT-N+S; all other modes are represented as too energetic in all OSSEs.

## 4 Discussion and Conclusions

Our exploration of the 3-dimensional representation of subsurface conditions in DA simulations has shown that even in the presence of high resolution subsurface temperature observations, subsurface dynamics are not being captured correctly. Despite several observations in the shallow waters, the thermocline and mixed layer are represented as deeper in the OSSEs compared to the Ref state (compare Fig. 2 first column to following columns). Likewise, in the deeper water between 250–2000 m, all OSSEs overestimate the time-mean EKE, with the vertical profiles of mean EKE in the high eddy variability region all showing an overestimation of EKE (Fig. 3k). Only with the assimilation of XBT observations near to the high eddy variability region does the model produce a reasonable estimate the mean EKE at depth (Fig. 3i). The overly steep $5°C$ isotherms (e.g. Fig. 5g-j) are another indication that baroclinic dynamics are not being represented correctly.

Exploring the case study eddies, it is clear that the presence of subsurface observations improves the representation of the thermal structure and baroclinic modes in the vicinity of those observations (e.g. Fig. 6i and Fig. 9b-d). However, at distance from those observations ($\sim$300 km, case study B; Fig. 7g-j), 3-dimensional representation is again poor. This suggests that spatially and/or temporally sparse observing platforms (i.e. Argo floats, quarterly XBT observations and sporadic glider deployments), likely do not help DA simulations to resolve the correct eddy structure, especially if are not directly co-located. The differences in the mode structures between the OSSEs and Ref state, and between the OSSEs with observations close to the eddies, show that: the primary thermocline slope is particularly susceptible to inaccuracy (see poor $\phi_2$ structure in XBT-N; Fig 10c); and, if there is secondary structure such as steps in the thermocline (i.e. a complex density structure; Fig. 9e-h), DA simulations will potentially struggle to generate representative baroclinic mode structure.

We have focussed on two ways in which the DA system may be impacting the dynamics of the vertical structure. The first is that baroclinic instability is too active, as a result of a poor vertical density structure. This is displayed in the baroclinic conversion rate for all OSSEs being higher than the Ref state (Fig. 8g-j), while the barotropic conversion rate is represented relatively well (cf. Fig. 8a to Fig. 8b-e). In the XBT OSSEs simulated here, the subsurface observations have fine horizontal and vertical spacing, which improves the vertical temperature structure (see, for example, better across-shelf slope in the XBT-S experiment in the vicinity of the observations; Fig. 5i). However, the improvement from these XBT observations does not extend far from the observation location.

The second manifestation of how dynamics are impacted from the DA process is through incorrectly representing the energy flow pathways and distribution through the baroclinic modes, leading to incorrect vertical structure. The presence of observations improves the baroclinic mode structure, as displayed by the XBT-S OSSE in Fig. 9b-d. However, this structure is degraded further from the observations (e.g. Fig. 9f-h), or in general in the presence of eddies, which have more complex vertical structure and are thus harder for the model dynamics to capture.

The poor subsurface representation in some DA simulations, including these experiments, may be due to a suboptimally specified background error covariance matrix. The background error covariance matrix is critical for performing data assimilation: it is used to weight the importance of the model forecast during the mathematical combination of model state and observations (Lee and Huang, 2020); it determines how observations exert influence in the vertical and horizontal directions

(Bannister, 2008a); and it describes correlations and synergies between observations (Bannister, 2008b). However, it is computationally unfeasible to explicitly set, compute or store this term (due to the large number of elements), and thus it must be estimated or modelled (e.g. Bannister, 2008a). In Ensemble DA, statistical methods are applied to an ensemble of forecast simulations to produce a background error covariance. This estimate can be iteratively updated when a new ensemble is available (and thus can evolve in time) and, as it is calculated from model output, it can contain different horizontal and vertical length scales (e.g. Brassington et al., 2007; Oke et al., 2008). However, due to the statistical nature of ensemble-based background error covariance estimates, large modes of variability will be dominant and smaller-scale components can be lost (Li et al., 2015). In Variational DA, the background error covariance must be estimated with a model (e.g. Weaver and Courtier, 2001) and often with assumptions of isotropy (similar horizontal and vertical length scales) and stationarity (no explicit flow-dependence) being made. The specification of the background error covariance matrix is indeed one of the biggest remaining challenges in development of DA and, in particular, 4DVar (Moore et al., 2019).

Most recent advances in improving estimates of this matrix stem from numerical weather prediction. For example, in weather prediction research, estimates of the background error covariance matrix have been investigated for various regions (Bonavita et al., 2011; Michel and Auligné, 2010; Lee and Huang, 2020), and there is continuing development in advanced assimilation schemes such as hybrid approaches that combine ensemble covariance statistics with static, 'climatological' covariance estimates (e.g. Lorenc and Jardak, 2018). These hybrid methods have the benefit of representing flow-dependent changes in the background error covariance from the ensemble covariance estimate, while counteracting the sampling noise inherent in the ensemble statistics with the static covariance estimate (Bonavita et al., 2011). Development of ocean DA techniques typically lags behind that of weather prediction. Some recent advances have been made in hybrid approaches, which, like their counterparts in weather prediction, combine an ensemble of model runs to estimate the background error covariance, which is then combined with a 4DVar scheme (e.g. Penny et al., 2015). Other research has focussed on modifying the DA algorithm such that covariance estimates can account for different spatial scales and model resolutions (e.g. Li et al., 2015). However, most present-day ocean DA systems do not apply sophisticated methods for estimating the background error covariance. As a result, it is possible that the background error covariance is, at least in part, responsible for the poor subsurface representation of dynamic and poorly sampled features (e.g. eddies) that we show here.

It is generally thought that poor background error covariance specification is a major impediment to improved ocean data assimilating simulations, which we hypothesise is the source of the aforementioned poor vertical structure. However, there is still much work to be done in this area. This future research could focus on improvements to estimates of the background error covariance, potentially using hybrid schemes or multi-scale approaches, or other aspects of DA schemes. Given the obvious motivation to improve the vertical representation of stratification and structure in eddies, there is justification for the continued development of basic research in ocean DA. In light of this, ocean DA can borrow much from developments and improvements in numerical weather prediction research.

*Code and data availability.* The exact version of the model, configuration files and forcing files used to produce the results used in this paper are archived on Zenodo (https://doi.org/10.5281/zenodo.6804480) and the UNSW library (https://doi.org/10.26190/unsworks/24146). The code used to conduct the normal mode analysis is available from https://doi.org/10.5281/zenodo.6999169.

## Appendix A: The 'baseline': Bias in the OSSE configuration

As described in Section 2.2, we employ a fraternal twin approach, where the ref state and the OSSE are simulated by the same model, but with different configurations. These differences, such as parameterisations and boundary conditions, should produce errors that are similar in nature (i.e. have similar magnitude and properties) to the initialisation error present in a true ocean DA system. However, the errors introduced through differences in configuration should not result in such a large impact, that the long-term representation is no longer realistic. If this occurs, it is difficult to separate out the error resulting from the difference in configuration (the bias), and what is the difference resulting from the DA process itself. Consequently, the free-running and data-assimilating simulations must have different configurations but without a large mean bias.

To quantify this bias, we run a 'baseline' experiment, using the free-running model but with boundary conditions and parameterisations identical to the OSSEs. The bias is then calculated as the time-mean difference between the ref state and the baseline simulation.

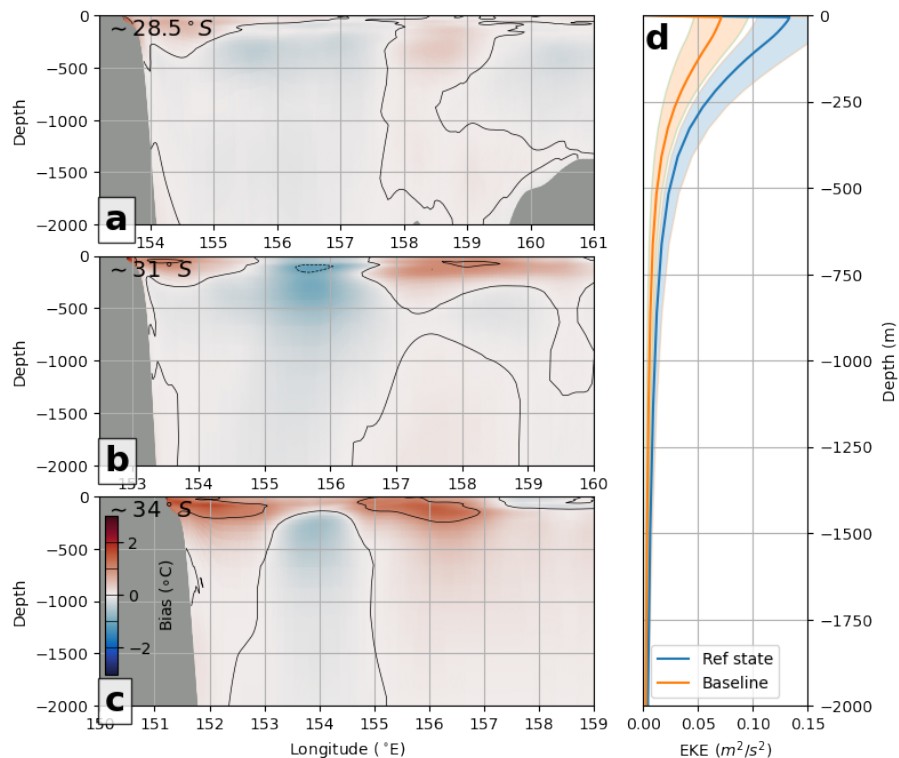

**Figure A1.** The temperature bias between the baseline and the ref state is shown at three transects, (a) ~28.5° S, (b) ~31° S and (c) ~34° S. In (d), the depth profile of EKE, averaged over the high EKE box (see box in Fig. 3a), for the ref state and baseline experiments.

Fig. A1 shows the time-mean bias in temperature at three transects: ~28.5° S, ~31° S, ~34° S (Fig. A1a-c). The surface region displays the greatest bias, of approximately 1.5°C in the surface waters at ~34° S (Fig. A1c), while at depth bias is negligible (close to 0°C below 500 m in all transects Fig. A1a-c). The surface bias is very likely to be corrected for by the assimilation of SST observations. The depth profile of EKE for the ref state and baseline have similar shape: surface intensified with a gradual decrease with depth. Compare this to the same profiles for the OSSEs, which display subsurface maxima (Fig 3k).

The lack of strong (subsurface) bias with a consistent sign suggests that the differences in subsurface structure (e.g. Fig. 2,4,5), mode structure (Fig. 9 and 10), EKE distribution (Fig. 3) and energy conversion rates (Fig. 8) are principally a product of the DA system; they don't result from any consistent bias in the DA model forcing and configuration.

## Appendix B: Example observation coverage

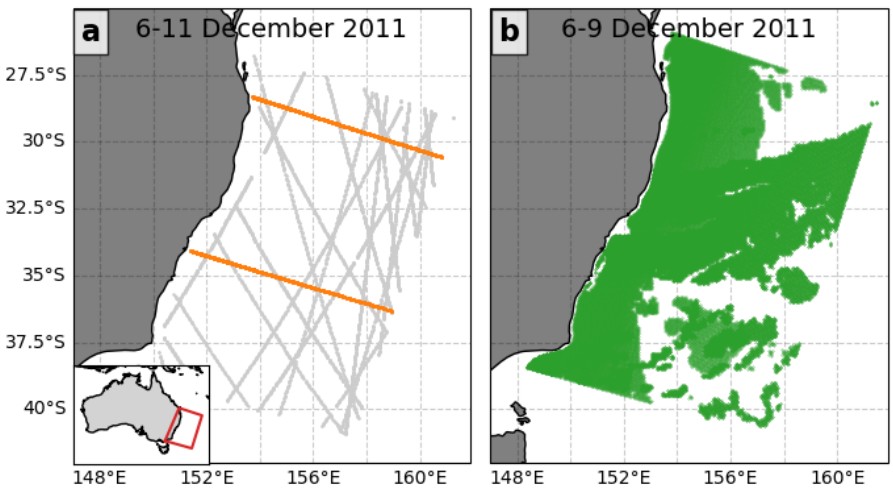

**Figure B1.** Example coverage of (a) along-track SSH (grey dots) and XBT (orange dots) over the period 6-11 December 2011, and (b) SST (green dots) over the period 6-9 December 2011. A shorter window is selected to show the typical spatial coverage of the SST, which, due to the high resolution and daily imaging, often covers the whole domain. Gaps in SST coverage are usually due to low surface winds or high cloudiness. These gaps are simulated using thresholds of $2 \mathrm{~m\,s^{-1}}$ and 0.75 (for low wind and cloudiness, respectively) and are applied to daily fields from the BARRA-R reanalysis. Methods for masking and preparation of the SST and other observations are given in detail in Gwyther et al. (2022).

## Appendix C: Model configuration details

Key configuration settings and differences between the ref state and the OSSE model configuration are shown in Table C1. The decorrelation length scales are set following Kerry et al. (2016), and are consistent with estimates used elsewhere (e.g. Zhang et al., 2010; Zavala-Garay et al., 2012; Kerry et al., 2018; Siripatana et al., 2020; Gwyther et al., 2022). Observation error covariances (see Table C1) are applied for each observation type. Further discussion of the preparation of the observations, the choice of error, and the minimization scheme is discussed further in Gwyther et al. (2022).

**Table C1.** Key differences in model configuration are shown between the Free-running ref state and the 4D-Var OSSEs. Further details are given in Gwyther et al. (2022) and references therein.

| Configuration | Free run | 4D-Var OSSE |
|---|---|---|
| Lateral BCs | BRAN2020 | BRAN2020 |
| Surface BCs | BARRA-R | ACCESS with bulk flux parameterisation |
| Mixing schemes | Harmonic horizontal mixing coefficient is $40\ \mathrm{m^2\,s^{-1}}$ for tracers and $55\ \mathrm{m^2\,s^{-1}}$ for momentum. Background vertical mixing coefficient is $1 \times 10^{-6}\ \mathrm{m^2\,s^{-1}}$ for tracers and $2 \times 10^{-5}\ \mathrm{m^2\,s^{-1}}$ for momentum. | Harmonic horizontal mixing coefficient is $200\ \mathrm{m^2\,s^{-1}}$ for tracers and $300\ \mathrm{m^2\,s^{-1}}$ for momentum. Background vertical mixing coefficient is $1 \times 10^{-6}\ \mathrm{m^2\,s^{-1}}$ for tracers and $1 \times 10^{-5}\ \mathrm{m^2\,s^{-1}}$ for momentum. |
| DA background error | n/a | Decorrelation length scales are assumed to be homogenous and isotropic. Horizontal length scale is $100\ \mathrm{km}$; Vertical length scale is $10\ \mathrm{m}$. |
| DA observation error | n/a | SSH error is $0.04\ \mathrm{m}$; SST error is $0.5^{\circ}\mathrm{C}$; XBT has a depth-varying error profile with a subsurface max of $0.6^{\circ}\mathrm{C}$ at $300\ \mathrm{m}$ decreasing to $0.12^{\circ}\mathrm{C}$ at $1100\ \mathrm{m}$. |
| DA 4D-Var loops | n/a | 14 inner loops and 1 outer loop. |
| More details | See Gwyther et al. (2022) and Li et al. (2021). | See Gwyther et al. (2022) and Kerry et al. (2016). |

*Author contributions.* DEG, CK, MR and SRK conceived and designed the experiments. DEG and CK performed the simulations. DEG and SRK analysed the data. All authors contributed to interpretation of the results. DEG prepared the paper with contributions from all co-authors.

*Competing interests.* The contact author has declared that neither they nor their co-authors have any competing interests.

*Acknowledgements.* We thank two anonymous reviewers for their comments, which helped to improve this work. This research and DG were supported by Australian Research Council Industry Linkage Grant LP170100498 to MR, SRK and CK. Former model development was supported by Australian Research Council grants DP140102337, LP160100162 to MR. This research was undertaken with the assistance of resources and services from the National Computational Infrastructure (NCI), under the grant *fu5*, as well as computations using the computational cluster Katana (https://doi.org/10.26190/669x-a286) supported by Research Technology Services at UNSW Sydney.

The current version of ROMS is available from the project website: https://www.myroms.org/projects/git-src/ under an open-source licence. We acknowledge that the forcing conditions are sourced from the Commonwealth Science and Industrial Research Organisation (BRAN2020; available at https://research.csiro.au/bluelink/outputs/data-access/) and the Bureau of Meteorology (BARRA-R and AC-

CESS; http://www.bom.gov.au/research/projects/). Along-track SSH data is sourced from the E.U. Copernicus Marine Service Information (https://doi.org/10.48670/moi-00146) and model configurations for the free-running and DA simulations are identical to those used in previous simulations (available online at https://doi.org/10.26190/TT1Q-NP46; https://doi.org/10.26190/5ebe1f389dd87). Our synthetic XBT data was modelled on XBT data made available by the Scripps High Resolution XBT program (www-hrx.ucsd.edu).

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
