# Peer review of "How does 4DVar data assimilation affect the vertical representation of mesoscale eddies? A case study with OSSEs using ROMS v3.9"

_Geoscientific Model Development, 2022_

## Author Comment (AC1)

**Response to reviewer comment RC1**

Reviewer comments are presented first in *blue italics*, then followed by the author's response in normal font. Line numbers are referring to the original manuscript and are denoted as L145 for Line 145. For changes to text, we include the original in red and the changed version or new additions in green.

*Dear Editor,*

*first of all I would like to say that I feel honored by the invitation to be a reviewer of the manuscript gmd-2022-204.*

*The manuscript entitled "How does 4DVar data assimilation affect the vertical representation of mesoscale eddies? A case study with OSSEs using ROMS v3.9" addresses a relevant and not entirely dominated question related to the quality of subsurface fields of ocean circulation models with observational data assimilation.*

*In general, the authors report their work in an organized, objective and well structured text, in a way that I could easily understand the problem, the methodological approach used, results obtained and what they indicate. The way the authors conduct the comparison of dynamic modes between solutions is particularly interesting.*

*I therefore recommend the publication of the paper after minor reviews are addressed.*

*Some questions, general comments, and suggested corrections are listed below so that the authors might want to address:*

**General comments:**

*line 18) wouldn't be "to deliver" instead of "the deliver"?*

Changed to "and, they deliver nutrients.." Thank you.

*line 90) although details of the DA model setup are reported in the other works cited, I missed some basic information of the 4D-Var implemented in ROMS that could somehow impact the results obtained, for example, the horizontal and vertical decorrelation scales, errors of the observations (table 1) and the number of inner and outer loops of the IS4D-Var.*

Following this comment and a comment of RC#2, we have added the following table to the Appendix.

**Table 1.** Key differences in model configuration are shown between the Free-running ref state and the 4D-Var OSSEs. Further details are given in Gwyther et al. (2022) and references therein.

| Configuration | Free run | 4D-Var OSSE |
| --- | --- | --- |
| Lateral BCs | BRAN2020 | BRAN2020 |
| Surface BCs | BARRA-R | ACCESS with bulk flux parameterisation |
| Mixing schemes | Harmonic horizontal mixing coefficient is $40\ \mathrm{m^2\,s^{-1}}$ for tracers and $55\ \mathrm{m^2\,s^{-1}}$ for momentum. Background vertical mixing coefficient is $1 \times 10^{-6}\ \mathrm{m^2\,s^{-1}}$ for tracers and $2 \times 10^{-5}\ \mathrm{m^2\,s^{-1}}$ for momentum. | Harmonic horizontal mixing coefficient is $200\ \mathrm{m^2\,s^{-1}}$ for tracers and $300\ \mathrm{m^2\,s^{-1}}$ for momentum. Background vertical mixing coefficient is $1 \times 10^{-6}\ \mathrm{m^2\,s^{-1}}$ for tracers and $1 \times 10^{-5}\ \mathrm{m^2\,s^{-1}}$ for momentum. |
| DA background error | n/a | Decorrelation length scales are assumed to be homogenous and isotropic. Horizontal length scale is 100 km; Vertical length scale is 10 m. |
| DA observation error | n/a | SSH error is 0.04 m; SST error is $0.5^{\circ}$C; XBT has a depth-varying error profile with a subsurface max of $0.6^{\circ}$C at 300 m decreasing to $0.12^{\circ}$C at 1100 m. |
| DA 4D-Var loops | n/a | 14 inner loops and 1 outer loop. |
| More details | See Gwyther et al. (2022) and Li et al. (2021). | See Gwyther et al. (2022) and Kerry et al. (2016). |

We have add the following information about the initial perturbation of the OSSEs:

"The OSSE that is simulating the same period as the Ref state is perturbed to introduce error and initiate divergent evolution (see discussion below)."

To

"The OSSE that is simulating the same period as the Ref state is perturbed to introduce error and initiate divergent evolution through the use of different initial conditions. These initial conditions are similar to those used to initialise the Ref state but are extracted from a point 8 days later (the OSSE begins at 2 December 2011 with conditions from 10 December 2011). This offset is chosen so as to fairly test the DA system (see Gwyther et al., 2022 for further information about this choice of perturbation)."

We have also included the following paragraph in the associated Appendix section, which discusses some of the differences between the model configurations.

"Key configuration settings and differences between the ref state and the OSSE model configuration are shown in Table 1. The decorrelation length scales are set following Kerry et al. (2016; section 3.5), and are consistent with estimates used elsewhere (e.g. Zhang et al., 2010; Zavala-Garay et al., 2012; Kerry et al 2018; Siripatana et al 2020; Gwyther et al., 2022)). Observation error covariances (see Table 1) are applied for each observation type. Further discussion of the preparation of the observations, the choices of error, and the minimization scheme is discussed further in Gwyther et al. (2022)."

*Has any sensitivity analysis been done in order to verify whether some of these assimilation parameters affect in a significant way the patterns found, i.e. the poor representation of the subsurface structures? Would reducing the errors of the observations bring the models closer to the reference state?*

We completely agree that a sensitivity study of parameters is important, in particular the sensitivity to the choice of how background error covariances are estimated. However, we believe this is firmly out of scope of this study. Here, we have focussed on how the oceanography in dynamic regimes is impacted by choices in observation strategies, rather than an exhaustive and technical exploration of DA system choices. The latter would likely require many new experiments in order to be a thorough analysis of the full range of parameter choices and would change the scope of the paper completely. (see also response to Reviewer 2).

With regards to the option of reducing observational error. Reducing the observational error may bring the estimates closer to the ref state/observations, however it may also 'overfit' the values leading to increased misfit elsewhere. We selected these values as they are the same or similar values to those used in other real (i.e. non-OSSE) data assimilating models. Furthermore, the synthetic observations are perturbed with errors that are normally distributed with a variance corresponding to the specified observation error, so the specified observations errors are 'correct' by definition.

*line 117) do the fact that the reference run has a distinct setup with different boundary conditions and vertical mixing schemes, for example, interfere with the ability of the assimilative run to converge to the reference solution with respect to, for example, mixed layer depth? could a distinct setup between the reference and assimilative model lead to biases that could not be corrected through DA? How do the reference simulation and the free integration of DA setup compare?*

Thank you for this interesting question. We chose to perform a 'fraternal twin'-type OSSE, which can be run with different model configurations between the ref state and the OSSE, or with the same model type between the ref state and the OSSE, but with a variety of other configuration differences. This is a deliberate choice so that the growth of errors results from several sources, as opposed to just initialisation error, which would be the case if we used identical model configuration. This is more appropriate for the simulation of a realistic data assimilation system, which includes errors from a variety of sources (e.g. numerical truncation error, initialisation error, errors in resolved and parameterised processes, and errors in boundary conditions). A good overview is given in Halliwell et al., (2014).

Given that we're also interested in subsurface representation, we don't want to introduce too much error for the DA system, such that it cannot correct for such large differences. That is why we chose to keep many model configuration parameters and forcings the same, except for the surface forcing and some mixing parameters (see new table for more details).

We acknowledge that we omitted a demonstration of a free-running integration of the data-assimilating configuration. We now show this below. This 'baseline' run shows the bias in the integration resulting from the different surface forcing and mixing parameters.

[Figure]

Figure A1. The temperature bias between the baseline and the ref state is shown at three transects, (a) 28.5S, (b) 31S and (c) 34S. In (d), the depth profile of EKE, averaged over the high EKE box (see box in Figure 3a), for the ref state and baseline experiments.

To explain this figure, a new section has been added to the appendix.

Appendix A: The `baseline': Bias in the OSSE configuration

As described in Section 2.2, we employ a fraternal twin approach, where the ref state and the OSSE are simulated by the same model, but with different configurations. These differences, such as parameterisations and boundary conditions, should produce errors that are similar in nature (i.e. have similar magnitude and properties) to the initialisation error present in a true ocean DA system. However, the errors introduced through differences in configuration should not result in such a large impact, that the long-term representation is no longer realistic. If this occurs, it is difficult to separate out the error resulting from the difference in configuration (the bias), and what is the difference resulting from the DA process itself. Consequently, the free-running and data-assimilating simulations must have different configurations but without a large mean bias.

To quantify this bias, we run a 'baseline' experiment, using the free-running model with boundary conditions and parameterisations identical to the OSSEs. The bias is then calculated as the time-mean difference between the ref state and the baseline simulation.

Figure A1 shows the time-mean bias in temperature at three transects: 28, 31, 34 (Figure A1a-c). The surface region displays the greatest bias, of approximately 1.5C in the surface waters at 34S (Figure A1c), while at depth bias is negligible (close to 0C below 500m in all transects Figure A1a-c). The surface bias is very likely to be corrected for by the assimilation of SST observations. The depth profile of EKE for the ref state and baseline have similar

shape: surface intensified with a gradual decrease with depth. Compare this to the same profiles for the OSSEs, which display subsurface maxima (Figure 3k).

The lack of strong (subsurface) bias with a consistent sign suggests that the differences in subsurface structure (e.g. Figure 2,4,5), mode structure (Figures 9 and 10), EKE distribution (Figure 3) and energy conversion rates (Figure 8) are principally a product of the DA system; they don't result from any consistent bias in the DA model forcing and configuration.

This appendix has also been introduced in the methods section:

"However, it is also important to ensure that the different configuration of the Ref state and OSSEs (e.g. in this case, surface forcing and some mixing parameters) do not cause such an impact as to introduce a large long-term bias. To assess this, a `baseline' experiment was conducted using the OSSE configuration, but without assimilating any observations. Comparison of this against the Ref state showed a warm bias in the surface waters, which is likely to be corrected by assimilating SST. More importantly, there is no strong bias in the subsurface ocean, which would otherwise be difficult to correct with assimilation (see Fig.A1)."

*line 167) how was the density perturbation rho' estimated? was it calculated as the perturbation of a time and area average density rho(z)?*

It was calculated as the difference from the time-mean density at each location, not an area average. This information has been added at L167: "are the density perturbation and vertical velocity perturbation from the long-term means calculated at each location, respectively,"

*Figure 5) fontsize of vertical axis ticklabels and titles are too small*

Thank you for pointing this out. The figure has been adjusted to have a smaller width and bigger font sizes. The same changes were also applied to Figure 4.

*Figure 6 and 7) colorbar labels are difficult to read on panels f) and I think a colorbar is missing for panels k) (green to red diverging colormap)*

Thank you for pointing this out - we have corrected both of these figures.

*line 340) is the word "sim" a typo in "sim350m"?*

Yes, we have corrected it to $\sim$ or the ~ symbol. Thank you.

*line 391) in this case, could the balance operator implemented in ROMS be favorable?*

As described in Kerry et al 2016, in this work, we only prescribe univariate covariance. The dynamics are coupled through the use of the tangent-linear and adjoint models in the assimilation, but not in the statistics of P (that is the matrix is a diagonal matrix and does not include balanced operators). The 4DVar balance operator may be a useful approach, however we have not used it and so could not comment on whether it will produce a large improvement. However we thank the reviewer for this suggestion, and will explore it in future work.

**Other changes:**

L412: Removed the first full stop in "static covariance estimate.(Bonavita et al., 2011)."

Figure 9: We have modified this plot to show a single mode in each column (instead of all modes in each OSSE per column). We believe this better demonstrates the failure of the OSSEs to represent each mode, especially without nearby observations. We also added the RMS difference from the Ref state for each mode in each OSSE. This new figure is shown below. The caption and figure description will also be adjusted to reflect this change.

[Figure]

---

## Author Comment (AC2)

**Response to reviewer comment RC2**

Reviewer comments are presented first in *blue italics*, then followed by the author's response in normal font. Line numbers are referring to the original manuscript and are denoted as L145 for Line 145. For changes to text, we include the original in red and the changed version or new additions in green.

**Major comments:**

*1. 1 L101- 102 "Further details of the free-running and DA configuration used in these OSSEs are given in Gwyther et al. (2022)." However, not much information has been provided in this present paper. It is simple to cite past work! However, such details, even in a summary format are very relevant to this paper, a table of differences between `ref' and `DA' model configurations would be most illustrative. Also a discussion of why those differences actually yield a "good" set-up to address the sub-surface impacts of data assimilation (discussed in the Introduction) is needed.*

Following this comment, we have added the following table to the Appendix. This table shows the key differences between the ref state and OSSE configurations and points the reader to references which explain the model setup in greater detail.

**Table 1.** Key differences in model configuration are shown between the Free-running ref state and the 4D-Var OSSEs. Further details are given in Gwyther et al. (2022) and references therein.

| Configuration | Free run | 4D-Var OSSE |
|---|---|---|
| Lateral BCs | BRAN2020 | BRAN2020 |
| Surface BCs | BARRA-R | ACCESS with bulk flux parameterisation |
| Mixing schemes | Harmonic horizontal mixing coefficient is 40 $m^2 s^{-1}$ for tracers and 55 $m^2 s^{-1}$ for momentum. Background vertical mixing coefficient is $1 \times 10^{-6} m^2 s^{-1}$ for tracers and $2 \times 10^{-5} m^2 s^{-1}$ for momentum. | Harmonic horizontal mixing coefficient is 200 $m^2 s^{-1}$ for tracers and 300 $m^2 s^{-1}$ for momentum. Background vertical mixing coefficient is $1 \times 10^{-6} m^2 s^{-1}$ for tracers and $1 \times 10^{-5} m^2 s^{-1}$ for momentum. |
| DA background error | n/a | Decorrelation length scales are assumed to be homogenous and isotropic. Horizontal length scale is 100 km; Vertical length scale is 10 m. |
| DA observation error | n/a | SSH error is 0.04 m; SST error is 0.5°C; XBT has a depth-varying error profile with a subsurface max of 0.6°C at 300 m decreasing to 0.12°C at 1100 m. |
| DA 4D-Var loops | n/a | 14 inner loops and 1 outer loop. |
| More details | See Gwyther et al. (2022) and Li et al. (2021). | See Gwyther et al. (2022) and Kerry et al. (2016). |

We have add the following information about the initial perturbation of the OSSEs:

"The OSSE that is simulating the same period as the Ref state is perturbed to introduce error and initiate divergent evolution (see discussion below)."

To

"The OSSE that is simulating the same period as the Ref state is perturbed to introduce error and initiate divergent evolution through the use of different initial conditions. These initial conditions are similar to those used to initialise the Ref state but are extracted from a point 8 days later (the OSSE begins at 2 December 2011 with conditions from 10 December 2011).

This offset is chosen so as to fairly test the DA system (see Gwyther et al., 2022 for further information about this choice of perturbation)."

We have also added the following information, emphasising differences between the Ref state and the OSSE configurations, by changing:

"The DA configuration uses lateral forcing conditions from BRAN2020 and surface forcing conditions from a bulk flux formulation (Fairall et al., 1996) with daily atmospheric conditions from the Australian Bureau of Meteorology's ACCESS reanalysis (Puri et al., 2013). The different surface forcing conditions between the DA configuration and the free-running Ref state are appropriate, as they lead to an additional source of error that the DA system must reduce."

To:

"The DA configuration uses lateral forcing conditions from BRAN2020 and surface forcing conditions from a bulk flux formulation (Fairall et al., 1996) with daily atmospheric conditions from the Australian Bureau of Meteorology's ACCESS reanalysis (Puri et al., 2013). Vertical and horizontal mixing parameters have also been modified between the free-run and data-assimilating configurations. The different surface forcing conditions and mixing parameters in the assimilating and free-running configurations (see Table 1) are appropriate, as they lead to a source of error that the DA system must reduce, as required in an OSSE."

Further we have included the following discussion in the associated Appendix section:

"Key configuration settings and differences between the ref state and the OSSE model configuration are shown in Table 1. The decorrelation length scales are set following Kerry et al. (2016; section 3.5), and are consistent with estimates used elsewhere (e.g. Zhang et al., 2010; Zavala-Garay et al., 2012; Kerry et al 2018; Siripatana et al 2020; Gwyther et al., 2022). Observation error covariances (see Table 1) are applied for each observation type. Further discussion of the preparation of the observations, the choices of error, and the minimization scheme is discussed further in Gwyther et al. (2022)."

With regards to demonstrating that this DA setup is a 'good' setup: All of the choices made in configuring the DA system were made following extensive research. This DA system has also been shown to produce accurate estimates and forecasts of the EAC when assimilating observations (as shown in Kerry et al., 2016 and subsequent publications e.g. Kerry et al., 2018; Siripatana et al., 2020). However, we could emphasise that further, and so we have added the following to L100, by changing:

"Further details of the free-running and DA configuration used in these OSSEs are given in Gwyther et al (2022)."

To:

"The performance and configuration options of the DA system were extensively tested and were shown to produce relatively low error in estimates and forecasts of the EAC (Kerry et al, 2016). The system uses 14 inner loops with one outer loop, set following testing of how many loops were required to achieve acceptable reduction in the cost function (see Gwyther

et al,. 2020; Kerry et al., 2016). The background error covariances are static and computed by factorisation based on Weaver and Courtier 2001, as described in detail in Kerry et al 2016."

*1.2 Table 1 "Along-track satellite-observed sea surface height altimetry and sea surface temperature." A plot is needed here to show the data coverage.*

We have added a supplementary figure with example coverage of SST and SSH observations, as shown below:

[Figure]

Supplementary Figure 2. Example coverage of (a) along-track SSH (grey dots) and XBT (orange dots) over the period 6-11 December 2011, and (b) SST (green dots) over the period 6-9 December 2011. A shorter window is selected to show the typical spatial coverage of the SST, which, due to the high resolution and daily imaging, often covers the whole domain. Gaps in SST coverage are usually due to low surface winds or high cloudiness. These gaps are simulated using thresholds of 2m/s and 0.75 (for low wind and cloudiness, respectively) using daily fields from the BARRA-R reanalysis. Methods for masking and preparation of the SST and other observations are given in detail in Gwyther et al., (2022).

This figure is now referenced in the methods section:

"Example coverage from SSH, XBT and SST are shown in Supplementary Figure 2."

*1.3 Figures such as 2 and 3 are time-averaged. But given the eddying flow, the errors are expected to vary in time, therefore a time-series plot or Hovmoller plot of error (with respect to `ref') standard deviations would be best.*

At the suggestion of the reviewer we experimented with a time-varying versus a time-averaged plot. However, we believe that a time-varying plot is not insightful. This is

because the eddies in each different model are not exactly at the same location and time, the comparison with the truth (e.g. an RMS plot) can have a large difference even if the fields aren't that dynamically different (e.g. an eddy is slightly misaligned). We feel this would be misleading. Our focus here is the time mean differences in subsurface structure hence we have not changed the plot.

*1.4 L 223 "The XBT-N+S OSSE has a slightly higher EKE difference than XBT-N or XBT-S but performs better than Surf (cf. Fig.3j and Fig.3g)" Why so? The authors do not explain this counterintuitive result though more observations have been added in XBT-N+S than XBT-N or XBT-S.*

We have changed:

"The XBT-N+S OSSE has a slightly higher EKE difference than XBT-N or XBT-S but performs better than Surf (cf. Fig.3j and Fig.3g)."

To

"The XBT-N+S OSSE has a slightly higher EKE difference than XBT-N or XBT-S but performs better than Surf (cf. Fig.3j and Fig.3g). As discussed in Gwyther et al., (2022), the XBT-N+S OSSE sometimes displays higher error than the single XBT transect OSSEs, which is likely because the DA scheme is forced to minimise errors at both the northern and southern subsurface observation locations. This leads to a degraded fit to either observation transect individually. This has also been demonstrated by others, for example, Siripatana et al., (2020), who found that additional data streams (mooring data and HF radar currents) degraded representation of SSH and SST; and Zhang et al., (2010), who showed that assimilating HF radar currents increased the error in the subsurface temperature forecast."

*1.5 L377 "DA simulations will potentially struggle to generate representative baroclinic mode structure." The authors generalize their conclusions without demonstrating how does the model used for DA performs without any assimilation (see comment above 1.1). They haven't shown the nature of errors that this model has without any observations - unless those are described, how can one draw conclusions whether assimilating observations helps or not? It is important to design the experiments properly, by making sure the model used for DA is different from that to generate the `ref' trajectory.*

We acknowledge that we omitted to show a free-running configuration of the data-assimilating configuration. So, we have run the free-running model with the same forcing conditions and mixing parameters as the OSSE configuration. This 'baseline' run shows the bias in the integration resulting from the different surface forcing and mixing parameters. We have included a new figure and section in the Appendix.

[Figure]

Figure A1. The temperature bias between the baseline and the ref state is shown at three transects, (a) 28.5S, (b) 31S and (c) 34S. In (d), the depth profile of EKE, averaged over the high EKE box (see box in Figure 3a), for the ref state and baseline experiments.

To explain this figure, a new section has been added to the appendix.

"Appendix A: The `baseline': Bias in the OSSE configuration

As described in Section 2.2, we employ a fraternal twin approach, where the ref state and the OSSE are simulated by the same model, but with different configurations. These differences, such as parameterisations and boundary conditions, should produce errors that are similar in nature (i.e. have similar magnitude and properties) to the initialisation error present in a true ocean DA system. However, the errors introduced through differences in configuration should not result in such a large impact, that the long-term representation is no longer realistic. If this occurs, it is difficult to separate out the error resulting from the difference in configuration (the bias), and what is the difference resulting from the DA process itself. Consequently, the free-running and data-assimilating simulations must have different configurations but without a large mean bias.

To quantify this bias, we run a 'baseline' experiment, using the free-running model but with boundary conditions and parameterisations identical to the OSSEs. The bias is then calculated as the time-mean difference between the ref state and the baseline simulation.

Figure A1 shows the time-mean bias in temperature at three transects: 28, 31, 34 (Figure A1a-c). The surface region displays the greatest bias, of approximately 1.5C in the surface waters at 34S (Figure A1c), while at depth bias is negligible (close to 0C below 500m in all transects Figure A1a-c). The surface bias is very likely to be corrected for by the assimilation

of SST observations. The depth profile of EKE for the ref state and baseline have similar shape: surface intensified with a gradual decrease with depth. Compare this to the same profiles for the OSSEs, which display subsurface maxima (Figure 3k).

The lack of strong (subsurface) bias with a consistent sign suggests that the differences in subsurface structure (e.g. Figure 2,4,5), mode structure (Figures 9 and 10), EKE distribution (Figure 3) and energy conversion rates (Figure 8) are principally a product of the DA system; they don't result from any consistent bias in the DA model forcing and configuration."

This appendix has also been introduced in the methods section:

"However, it is also important to ensure that the different configuration of the Ref state and OSSEs (e.g. in this case, surface forcing and some mixing parameters) do not cause such an impact as to introduce a large long-term bias. To assess this, a `baseline' experiment was conducted using the OSSE configuration, but without assimilating any observations. Comparison of this against the Ref state showed a warm bias in the surface waters, which is likely to be corrected by assimilating SST. More importantly, there is no strong bias in the subsurface ocean, which would otherwise be difficult to correct with assimilation (see Fig.A1)."

*1.6 I enjoyed reading Sec 2.3.4 and 3.5; other sections/results identified problems with the data assimilation experiments, but never explained their causes. In the end, I am not what exactly sure what is the take home message? Every data assimilation scheme/implementation has a specific treatment of background errors. In this paper, there is no concrete evidence that it is the background errors that are to be blamed; mere speculations have been raised (L391, L418). It would be much better if there was a set of experiments with different B (formulation or changed values) that proved these speculations-even partly. Otherwise, what exactly is the contribution of this work? A diagnostic tool presented in Sec 2.3.4 and its use case in Sec 3.5? I hope some of this criticism helps improve this work on a very important topic.*

We agree that a sensitivity study of DA parameters, like the background error covariance, would be interesting and very useful. To conduct such a sensitivity study, many new experiments would need to be simulated in order to thoroughly explore the parameter space and sensitivities of the parameter. This would completely change the scope of the paper. This manuscript is a case study of a particular DA configuration, where the goal is to assess in detail the subsurface representation of the EAC and its eddy field. We have demonstrated that assimilation of the different datasets changes the sub-surface structure of the eddies - which is often overlooked in the usage of data assimilating models. Indeed, few people look below the surface at dynamical features at all.

The goal of this paper is not to optimise (or improve) the representation of the background error covariances, but to assess the impact of hypothetical observing platforms and show how subsurface representation is impacted by data assimilation. However, we thank the reviewer for these comments - and we think they will be valuable for designing future research directions. Indeed, this study motivates our future research direction in which we

hope to improve the specification of the background error covariances using hybrid Ensemble-Var methods, as discussed at lines 406-412.

**Minor comments:**

*2. 1 L19 "they" deliver*

Changed.

*2.2 L68 "focus on two" ??*

Changed to "focus on two manifestations of this impact:"

*2.3 L340 "sim" should be "$\sim$"*

Changed

*2.4 Most figures do not have axes labeled (for e.g., "Latitude (deg N)"), same remark for colorbar - Fig.2*

We have ensured that all latitude and longitude axis labels are marked with a °S or °E suffix. We hope this should negate the need to include the axis label, however we have also added to every caption which has this: "Axes with latitudes are labelled °S and axes with longitudes are labelled °E"

We have also added colorbar labels to Figure 2, and removed a degree symbol in the caption.

*2.5 Most figures would be more readable if XBT-N, -S lines are superimposed.*

Thank you for suggesting this. We have updated all spatial maps to have XBT-N and XBT-S lines in the appropriate panels.

*2.6 L87 "surface forcing conditions from BARRA-R" Are they also daily?*

Correct. Have changed to "daily surface forcing conditions from BARRA-R"

*2.7 a. Fig.6(e) The transect line is further from XBT-S than in other panels. Is it same section?*

As we describe at L270-272, the transect line has been shifted so as to pass through the centre of the eddy, which should allow for a more comparison between OSSEs when eddies may have a slightly different position. Otherwise the comparison would be between the centre of the ref state eddy and the edge of the OSSE eddy.

*2.7 b. Fig 6(k), Fig. 7(k) What is the colorscale?*

Thank you for pointing this out - we have corrected both of these figures.